# Probabilistic Tree-of-thought Reasoning for Answering Knowledge-intensive Complex Questions

**Shulin Cao**[1*], **Jiajie Zhang**[1*], **Jiaxin Shi**[2], **Xin Lv**[1], **Zijun Yao**[1], **Qi Tian**[2†],
**Juanzi Li**[1], **Lei Hou**[1]

[1]Department of Computer Science and Technology, Tsinghua University, Beijing, China
[2]Cloud BU, Huawei Technologies

{caosl19, jiajie-z19}@mails.tsinghua.edu.cn
tian.qi1@huawei.com, {houlei, lijuanzi}@tsinghua.edu.cn

## Abstract

Large language models (LLMs) are capable of answering knowledge-intensive complex questions with chain-of-thought (CoT) reasoning. However, they tend to generate factually incorrect reasoning steps when the required knowledge is not available or up-to-date in models' parameters. Recent works turn to retrieving external knowledge to augment CoT reasoning. Despite being promising, these chain-based methods suffer from: 1) Negative retrieval. Unnecessary or incorrect retrieval may mislead the reasoning; 2) Limited sight. Lacking the ability to look backward or forward, a local error in one step will propagate along the chain.

In this paper, we propose a novel approach: **Probabilistic Tree-of-thought Reasoning (ProbTree)**. First, LLMs translate a complex question into a query tree, in which each non-root node denotes a sub-question of its parent node. Then, probabilistic reasoning is conducted over the tree, by solving questions from leaf to root considering the confidence of both question decomposing and answering. During reasoning, for leaf nodes, LLMs choose a more confident answer from *Closed-book QA* that employs parametric knowledge and *Open-book QA* that employs retrieved external knowledge, thus eliminating the negative retrieval problem. For non-leaf nodes, with the hierarchical structure, LLMs have broader sights and are able to globally reason with the information from child nodes, thus recovering from local errors. The experiments on three Complex QA datasets under the open-domain setting show that our approach outperforms SOTA methods significantly, demonstrating the effect of probabilistic tree-of-thought reasoning.

## 1 Introduction

Answering knowledge-intensive complex questions requires reasoning over multiple knowledge facts

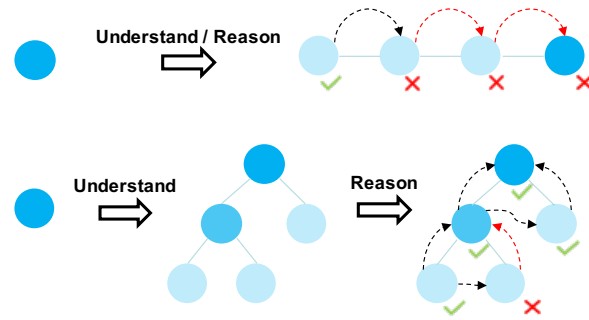

Figure 1: In chain-of-thought reasoning, the error in one step propagates to the final answer due to limited sight. Tree-of-thought reasoning can eliminate this problem by reconsidering the answer on the non-leaf nodes. The red edge means error propagation.

to infer the answer, and involves various reasoning capabilities such as multi-hop inference, attribute comparison, and set operation (Yang et al., 2018; Trivedi et al., 2022b; Cao et al., 2022). Large language models (LLMs) (Brown et al., 2020; Ouyang et al., 2022; Chowdhery et al., 2022) have shown to be capable of this task by breaking questions into a sequence of reasoning steps, termed chain-of-thought (CoT), before arriving at a final answer (Wei et al., 2022; Kojima et al., 2022). However, they tend to generate factually incorrect reasoning steps when the required knowledge is not available or up-to-date in their parameters (Trivedi et al., 2022a; Yao et al., 2022).

To mitigate this problem, retrieval-augmented LLMs have attracted increasing attention (Shi et al., 2023; Jiang et al., 2023), which retrieve knowledge from an external datastore when needed and then perform CoT reasoning conditioning on the retrieved knowledge. Typically, they retrieve multiple times in an iterative way, using the last generated sub-question (Press et al., 2022) or reasoning step (Trivedi et al., 2022a) as the query to retrieve documents and generate the next one based on it.

Despite leading to performance gains, these chain-based methods are still unsatisfactory for two

---

*Indicates equal contribution.
†Corresponding author.

reasons: 1) Negative Retrieval. They ignore that unnecessary or incorrect external knowledge may mislead the LLM reasoning (Jiang et al., 2023). *E.g.*, we investigate the inference results of IR-CoT (Trivedi et al., 2022a) (one of the SOTA retrieval-augmented LLMs) , finding that around 10% of questions are incorrectly answered due to this reason; 2) Limited Sight. Lacking the ability to look forward and backward, a local error in one step will propagate along the chain, deteriorating the whole reasoning process. *E.g.*, as shown in Fig. 1, an incorrectly answered sub-question will cause the following generated sub-questions incorrect, causing the final answer incorrect.

To this end, inspired by previous works that conduct optimization on tree-like computation graphs (Bai et al., 2022; Liu et al., 2023; Zhang et al., 2023), in this paper, we propose a novel approach: **Probabilistic Tree-of-thought Reasoning (ProbTree)**. In all, we disentangle QA into two stages: understanding and reasoning. As shown in Fig. 1, first, LLMs translate a given question into a query tree via the language understanding ability. In this tree, the root node is the original complex question, and each non-root node is the sub-question of its parent. Each leaf node is an "atomic" question that cannot be further decomposed. Second, reasoning is conducted over the tree, by solving questions from leaf to root with post-order traversal, considering the confidence of both question decomposing and answering. Via ProbTree, we can alleviate the above two problems: 1) During reasoning, for leaf nodes, *Closed-book QA* that employs parametric knowledge and *Open-book QA* that reasons with retrieved external knowledge are conducted simultaneously, and LLMs choose a more confident answer from them based on self-evaluation, thus eliminating the negative retrieval problem; 2) On non-leaf nodes, LLMs are able to globally reason with the information from child nodes. With the hierarchical structure of tree, LLMs have broader sights and can recover from wrong decompositions or incorrectly answered sub-questions, thus alleviating the problem of error propagation.

Specifically, given the observation that LLMs tend to be well-calibrated and low probability often indicates a lack of knowledge (Jiang et al., 2021; Kadavath et al., 2022; Jiang et al., 2023), we hypothesize that likelihood of the explanation (*i.e.*, reasoning steps in CoT) indicates LLM confidence

in the answer, and propose to quantify answer confidence with explanation logits. During forward propagation, for each node, LLMs choose the most confident answer from three QA modules: 1) *Child-aggregating QA* that reasons with child question-answer pairs; 2) *Open-book QA* that reasons with the external knowledge including the retrieved paragraphs for itself and its descendants[1]; 3) *Closed-book QA* that employs the implicitly encoded parametric knowledge.

In evaluation, we instantiate ProbTree on three Complex QA datasets under the open-domain setting: HotpotQA (Yang et al., 2018), MusiQue (Trivedi et al., 2022b), and 2WikiMultihopQA (Ho et al., 2020). Experimental results using OpenAI GPT3 (`text-davinci-003`) show that ProbTree improves the performance significantly, by 3.9%, 7.3%, and 5.2% F1 score on the three datasets respectively compared with existing SOTA models.

**Our contributions** include: a) exploring the LLM capacity of answering knowledge-intensive complex questions and proposing the tree-of-thought reasoning approach for the first time; b) bringing uncertainty into reasoning and proposing a self-evaluation based method to quantify answer confidence, which enables integrating external and parametric knowledge in a unified framework; c) demonstrating the effect of ProbTree with in-depth analyses, and proving the effect of each component with careful ablation studies. Our codes and datasets can be obtained from `https://github.com/THU-KEG/ProbTree`.

## 2 Related Work

**Retrieval-augmented Large Language Models** For retrieval-augmented LLMs, previous works typically adopt one-time retrieval, *i.e.*, retrieve knowledge using only the input question once (Borgeaud et al., 2022; Izacard et al., 2022; Shi et al., 2023). This line of work, however, cannot meet the need of complex questions that require multi-hop inference. Recently, another line of work arose, which adopt multi-time retrieval during the generation process to meet the complex information needs. The representative works include SelfAsk (Press et al., 2022) which prompts LLMs to decompose a question into sub-questions and triggers retrieval

---

[1]For leaf nodes, the output from *Child-aggregating QA* is empty, and the external knowledge in *Open-book QA* only contains the retrieved paragraphs for itself.

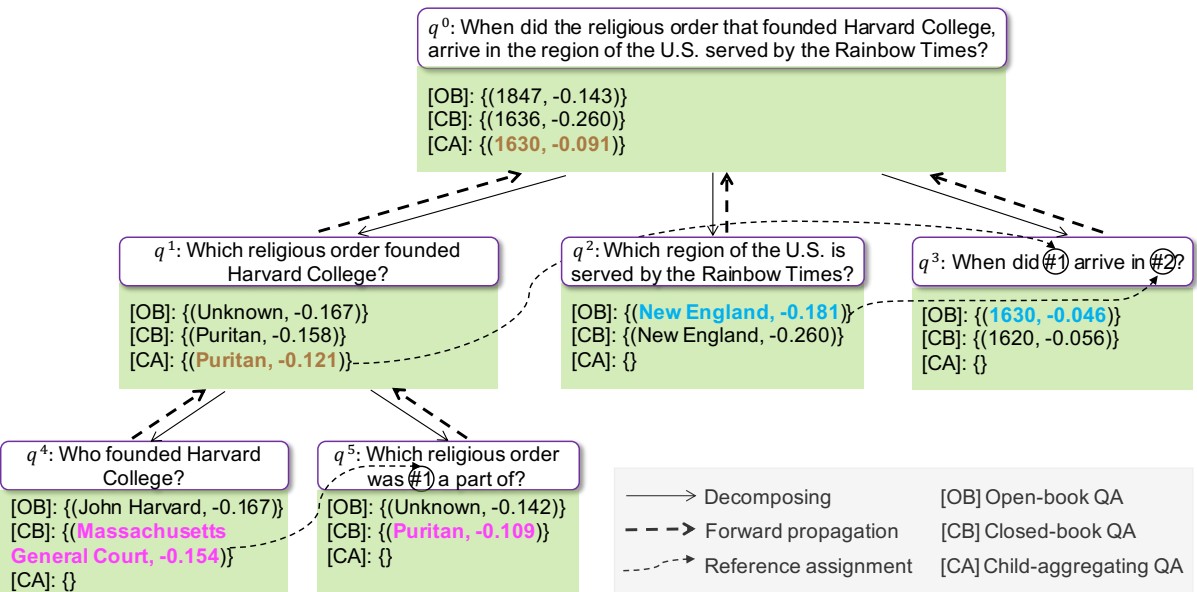

Figure 2: Illustration of the reasoning phase of ProbTree. In this phase, reasoning is conducted over the tree by solving questions from leaf to node via a forward propagation. For each node, LLMs choose the most confident answer from three QA modules: *Child-aggregating QA*, *Open-book QA*, and *Closed-book QA*.

every sub-question, IRCoT (Trivedi et al., 2022a) which triggers retrieval every CoT sentence (*i.e.*, reasoning step), ITER-RETGEN (Shao et al., 2023) which leverages the generation (the complete CoT reasoning steps) from the previous turn concatenated with the original question as the query for next turn generation, DSP (Khattab et al., 2022) which employs task-aware programs to decompose the question and perform retrieval to solve sub-questions, *etc.*

Compared with their works, we are the first to organize the complex information needs with a tree structure, which has broader sights and more flexibility. Besides, we are the first to bring uncertainty into reasoning for integrating parametric and external knowledge in a unifided famework.

**Reasoning on Tree-like Computation Graphs**
Tree is a basic data structure in computer science, and there have been works that employ tree structure to solve complex tasks in recent years. For example, Yao et al. (2023) prompts LLMs to perform BFS or DFS searching to solve tasks that need planning such as Game 24. Given a complex logical query or natural language question, several works derive a tree-like computation graph from the input, and then reason over the tree recursively to obtain the global solution (Bai et al., 2022; Liu et al., 2023; Zhang et al., 2023). The tree structure brings explainability to their model. However, they

rely heavily on supervision data which are expensive to obtain (*e.g.*, sub-question annotations), and need to call specially-trained sub-modules such as link predictors, neural readers, semantic parsers, *etc.* In contrast, our work turns to fully exploring the potentials of LLMs, *i.e.*, whether this methodology can help improve LLMs' ability.

## 3 Methodology

In this section, we first formalize the query tree, then overview our framework, and finally introduce the details of each component of the framework.

**Query Tree** Given a question $Q$, its query tree is $T$, in which the root node is the input question, and each non-root node is a sub-question of its parent node. Each leaf node is an "atomic" question that cannot be further decomposed. We index the nodes of $T$ with BFS ordering. For example, in Fig. 2, the root node $Q$ is indexed with $q^0$, and its child nodes are $\langle q^1, q^2, q^3 \rangle$. Formally, for a non-leaf node $q^i \in T$, $child^i$ contains its child nodes index, and the child nodes are: $q^i.children = \left\langle q^{child^i_1}, \cdots, q^{child^i_n} \right\rangle, n \leq 3$. For non-leaf node $q^i$, reference tokens indicating entity variables such as "#$k$" ($k < n$) may appear in its child question $q^{child^i_j}, j \geq 2$, referring to the answer of question $q^{child^i_k}$. For example, in Fig. 2, the "#1" in $q^5$ ($q^{child^1_2}$) means the answer of $q^4$ ($q^{child^1_1}$).

**Overview** The question answering task is disentangled into two phases: understanding and reasoning. First, the **understanding phase** (Section 3.1) transforms the user input question into a query tree with the language understanding ability of LLMs. At the same time, the confidence scores of question decomposing are calculated, which will be used in the next probabilistic **reasoning phase** (Section 3.2). As shown in Fig. 2, we bring uncertainty into reasoning by considering the answer confidence from different QA modules. Specifically, during reasoning, LLM try to solve questions from leaf to root via post-order traversal. *E.g.*, in Fig. 2, $\langle q^4, q^5, q^1, q^2, q^3, q^0 \rangle$ are solved in order. For questions that contain reference tokens, the reference tokens are first assigned with a concrete entity that is from a previous answer. *E.g.*, for $q^5$, "#1" refers to the answer of $q^4$ "Massachusetts General Court", and $q^5$ is reformed as "Which religious order was Massachusetts General Court a part of?" before answering. For question solving, for node $q^i$, the optimal answers for its child nodes have been obtained, and LLMs consider answering $q^i$ based on 1) the answers of its sub-questions (*Child-aggregating QA*); 2) the retrieved external knowledge (*Open-book QA*); and 3) the parametric knowledge (*Closed-book QA*). For each QA module, LLMs output the confidence in the answer, and take the answer with the highest confidence as the final optimal solution for $q^i$.

## 3.1 Understanding

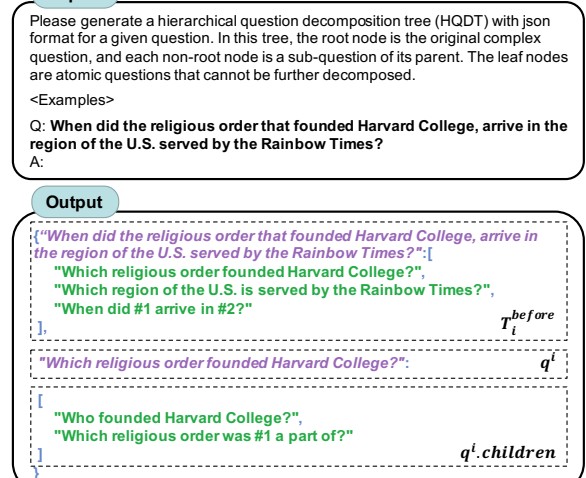

Figure 3: Given a question, LLMs generate the query tree with few-shot prompting. The query tree is in JSON format. The parent question is denoted as purple, and the list of child questions are denoted as green.

In this phase, given a question $Q$, its query tree $T$ is obtained, where $Q$ is the $q^0$ in $T$. In addition, for each non-leaf node $q^i \in T$, the decomposition score $ds^i$ is calculated to denote the confidence that $q^i$ can be decomposed into $q^i.children$.

Basically, as shown in Fig. 3, the query tree is generated by LLMs with few-shot prompting. The query tree is output in JSON format, where the key is the parent question, and the value is a list of child questions. A challenge here is to calculate the decomposing score. Given the observation that a generative pre-training model will assign a higher probability of high-quality generations than unqualified ones, we calculate $ds^i$ with the conditional probability of LLMs. Specifically, LLMs generate the query tree in BFS order, *i.e.,* all nodes present in the same level are generated first before generating the next level. Assume the generated tree before $q^i$ is denoted as $T_i^{before}$, and the serialized senquence of $q^i.children$ is $seq^i$. Suppose $seq^i = \langle x_1, \cdots, x_j, \cdots, x_{|seq^i|} \rangle$, $ds^i$ is calculated based on the average log-likelihood of $seq^i$:

$$ds^i = \frac{1}{|seq^i|} \sum_{j=1}^{|seq^i|} \log p(x_j | x_{<j}, [Q, T_i^{before}, q^i]). \tag{1}$$

Here $[\cdot, \cdot]$ means concatenation following the specified order.

## 3.2 Reasoning

In this phase, given $T$, $f_{qa}$ is conducted to find the optimal solution for $q^i$ from leaf to root on $T$ in post-order traversal,

$$f_{qa}(q^i, T, ds^i) \rightarrow (a^i, s^i). \tag{2}$$

Here, $a^i$ is the optimal answer for $q^i$, and $s^i$ is the corresponding confidence score.

In general, the optimal answer is the most confident answer from three QA modules: 1) *Child-aggregating QA* $f_{ca}$, 2) *Open-book QA* $f_{ob}$, and 3) *Closed-book QA* $f_{cb}$. Formally,

$$(a_{ca}^i, s_{ca}^i) = f_{ca}(q^i, T, ds^i), \tag{3}$$

$$(a_{ob}^i, s_{ob}^i) = f_{ob}(q^i, T), \tag{4}$$

$$(a_{cb}^i, s_{cb}^i) = f_{cb}(q^i), \tag{5}$$

$$m^* = \operatorname{argmax}_{m \in \{ca,\ ob,\ cb\}} s_m^i, \tag{6}$$

$$(a^i, s^i) = (a_{m^*}^i, s_{m^*}^i). \tag{7}$$

Here, $a_{ca}^i$, $a_{ob}^i$ and $a_{cb}^i$ are the candidate answers from the three QA modules respectively, and $s_{ca}^i$,

$s^i_{ob}$, $s^i_{cb}$ are the respective confidence scores. In the following, we will introduce the three QA modules in detail.

**Input**

Given a question and a context, answer the question and explain why.
<Examples>
Context:
Which religious order founded Harvard College? Puritan.
Which region of the U.S. is served by the Rainbow Times? New England.
When did Puritan arrive in New England? 1630.
Question:
**When did the religious order that founded Harvard College, arrive in the region of the U.S. served by the Rainbow Times?**
Answer:

**Output**

The religious order that founded Harvard College, the Puritans, arrived in the region of the U.S. served by the Rainbow Times, New England, in 1630.
So the answer is: 1630.

Figure 4: An illustration for the prompt of $f_{ca}$. The child question-answer pairs are denoted with green, and the explanation of answer is denoted with purple.

**Child-aggregating QA** As shown in Fig. 4, for a question, the question-answer pairs of its child nodes are packed into the prompt as its context, based on which LLMs perform CoT reasoning. This way, LLMs meta-reason with the information from child nodes, and give a comprehensive answer for the parent question. This is effective to recover from the wrong question decomposition or incorrectly answered sub-questions, helping with the error propagation due to limited sight in chain-based methods. Specifically, given a question $q^i$, $a^i_{ca}$ is outputed by the LLM ("1630" in Fig. 4). The challenge here is to obtain the confidence $s^i_{ca}$.

Intuitively, the likelihood of the explanation indicates the confidence of its derived answer. Therefore, suppose the context is denoted as $c$, and the explanation $e = \langle x_1, \cdots, x_j, \cdots, x_{|e|} \rangle$, the confidence $\tilde{s}^i_{ca}$ is calculated as:

$$\tilde{s}^i_{ca} = \frac{1}{|e|} \sum_{j=1}^{|e|} \log p(x_j | x_{<j}, [c, q^i]). \quad (8)$$

In addition, the correctness of $a^i_{ca}$ is also dependent on the correctness of its context, *i.e.*, the question-answer pair of its child nodes. Therefore, the final confidence score $s^i_{ca}$ is:

$$s^i_{ca} = \frac{1}{|n|+2} (ds^i + \sum_{j=1}^{|child^i|} s^{child^i_j} + \tilde{s}^i_{ca}). \quad (9)$$

**Open-book QA** The basic implementation of $f_{ob}$ is similar to that of $f_{ca}$, and $s^i_{ob}$ is also calculated with the CoT explanation logits (Eq. 8). The difference lies in the construction of context $c$.

Intuitively, for a question, the retrieved paragraphs for its descendants may contain the supporting evidence. Therefore, we take the related paragraphs $q^i.para$ as the context $c$:

$$q^i.para = R(q^i) \cup \bigcup_{j=1}^{|child^i|} q^{child^i_j}.para, \quad (10)$$

where $R(q^i)$ is the retrieved paragraphs with $q^i$ as query.

**Closed-book QA** For $f_{cb}$, the basic implementation is similar to that of $f_{ob}$, except that the context $c$ for the question $q^i$ is empty. $s^i_{cb}$ is also calculated with the CoT explanation logits (Eq. 8).

## 4 Experimental Settings

### 4.1 Datasets

We evaluate ProbTree on three Complex QA datasets under the open-domain setting: HotpotQA (Yang et al., 2018), MuSiQue (Trivedi et al., 2022b) and 2WikiMultiHopQA (2WikiMQA) (Ho et al., 2020). We use the same retrieval corpora as IRCoT (Trivedi et al., 2022a): for HotpotQA, we use the October 2017 Wikipedia dump; for MuSiQue and 2WikiMQA, they are constructed by combining all supporting and non-supporting paragraphs for all the questions in their train, dev, and test sets.

For each dataset, we also use the same development and test set provided by IRCoT (Trivedi et al., 2022a). Specifically, they randomly sampled 100 questions from the original dev set as the dev set, and another 500 as the test set.

### 4.2 Implementation Details

We instantiate ProbTree with OpenAI GPT3 (text-davinci-003) (Ouyang et al., 2022). The temperature is set to 0 for stable output.

Following IRCoT (Trivedi et al., 2022a), we use BM25 (Robertson and Zaragoza, 2009) implemented with Elasticsearch for the retriever $R$. For node $q^i \in T$, we retrieve top-$K$ paragraphs as $R(q^i)$, where $K \in \{3, 5, 7\}$ is selected based on the dev set.

| Method | HotpotQA | | | MuSiQue | | | | 2WikiMQA | | | | |
|---|---|---|---|---|---|---|---|---|---|---|---|---|
| | Overall | Bridge | Comparison | Overall | 2hop | 3hop | 4hop | Overall | Bridge | Inference | Comparison | Bridge-Comparison |
| Without Retrieval | | | | | | | | | | | | |
| Direct | 39.5 | 35.1 | 59.8 | 16.9 | 17.9 | 15.6 | 16.1 | 33.6 | 12.8 | 26.3 | 52.6 | 56.2 |
| CoT | 46.7 | 42.7 | 65.3 | 23.1 | 27.8 | 21.0 | 13.8 | 39.4 | 20.9 | 23.6 | 60.5 | 62.2 |
| With Retrieval | | | | | | | | | | | | |
| OneR | 53.2 | 50.0 | 68.4 | 25.7 | 31.0 | 22.6 | 16.0 | 48.1 | 18.6 | 29.8 | 86.6 | 73.7 |
| IRCoT | 60.2 | 58.0 | 69.4 | 34.2 | 44.2 | 26.3 | 20.1 | 63.8 | 46.2 | 45.4 | 91.6 | 79.0 |
| Self-Ask | 49.4 | 45.3 | 68.6 | 33.4 | 42.3 | 26.2 | 20.8 | 66.6 | 51.9 | 57.0 | 85.5 | 80.0 |
| ReAct | 44.7 | - | - | - | 37.0 | - | - | 38.5 | - | - | - | - |
| ITER-RETGEN | 61.1 | - | - | - | 42.0 | - | - | 48.1 | - | - | - | - |
| MCR | 57.0 | - | - | - | - | - | - | 67.9 | - | - | - | - |
| ProbTree | 62.6 (2.4 ↑) | 61.0 | 69.8 | 41.5 (7.3 ↑) | 50.1 | 38.1 | 23.3 | 71.8 (5.2 ↑) | 50.6 | 68.7 | 88.2 | 95.2 |
| | 64.1* (3.9 ↑) | 60.9* | 79.1* | | | | | | | | | |

Table 1: Answer F1 results of different methods. We highlight the best results in bold and second with an underline. * means we add the top-1 snippet from Google Search into $R(q^i)$ for leaf questions, via SerpAPI service (https://serpapi.com/) with the query format "en.wikipedia.org $q_i$". Color grey denotes the methods using different settings from ours, including GPT versions, retrievers and test samples. The results of ReAct and ITER-RETGEN are from (Shao et al., 2023), and the results of MCR are from their original paper.

## 4.3 Baselines

We mainly compare ProbTree with representative models that are in the same setting as us, including test set, document corpora, and retriever:

**Direct Prompting** prompts an LLM to directly predict an answer.

**CoT Prompting** prompts an LLM to generate step-by-step explanation before giving the answer.

**One-step Retrieval (OneR)** augments CoT Prompting by using the original complex question as a query to retrieve $K$ paragraphs. $K \in \{5, 10, 15\}$ is selected based on the dev set.

**IRCoT** (Trivedi et al., 2022a) interleaves retrieval with CoT generation by triggering retrieval every CoT sentence to generate the next one. We re-implement it with `text-davinci-003` (Details in Appendix C.1).

**Self-Ask** (Press et al., 2022) interleaves retrieval with follow-up sub-question generation by triggering retrieval every sub-question to generate the next one. We re-implement it with `text-davinci-003` (Details in Appendix C.2).

In addition, we include other recent approaches **ReAct** (Yao et al., 2022), **ITER-RETGEN** (Shao et al., 2023) and **MCR** (Yoran et al., 2023). Their results are not generally apples-to-apples comparisons, since they span a variety of evaluation settings, including GPT versions, test samples, and retrievers. Nonetheless, we report them here as qualitative reference points.

| Method | HotpotQA | MuSiQue | 2WikiMQA |
|---|---|---|---|
| ProbTree | **64.1** | **41.5** | **71.8** |
| SD | 51.7 (12.4 ↓) | 39.0 (2.5 ↓) | 69.2 (2.6 ↓) |
| w/o CA | 63.9 (0.2 ↓) | 34.8 (6.7 ↓) | 65.1 (6.7 ↓) |
| w/ RC | 53.2 (10.9 ↓) | 25.0 (16.5 ↓) | 52.0 (19.8 ↓) |
| w/ SC@3 | 62.4 (1.7 ↓) | 38.4 (3.1 ↓) | 68.8 (3.0 ↓) |
| w/ SC@5 | 63.0 (1.1 ↓) | 40.7 (0.8 ↓) | 70.1 (1.7 ↓) |
| w/o CB | 63.6 (0.5 ↓) | 38.4 (3.1 ↓) | 70.4 (1.4 ↓) |
| w/o OB | 46.3 (17.8 ↓) | 23.2 (18.3 ↓) | 42.3 (29.5 ↓) |

Table 2: Ablation results that demonstrate the effect of 1) Tree-of-thought reasoning: Sequential decomposition (SD), ProbTree without *Child-aggregating QA* (w/o CA); 2) Probilistic reasoning that integrates parametric and external knowledge in a unified framework: ProbTree with random choosing (w/ RC), with Self-consisency (w/ SC@3 and w/ SC@5), ProbTree without $f_{ob}$ (w/o OB), without $f_{cb}$ (w/o CB).

## 5 Experimental Results

### 5.1 Overall Results

As shown in Table 1, our method achieves the best performance on all three datasets. Compared with the SOTA method IRCoT, ours improves the F1 score by 2.4%, 7.3%, and 8.0% (5.2% compared with Self-Ask) respectively, demonstrating the effect of ProbTree. We attribute the improvement to three reasons: a) IRCoT suffers from the problem of error propagation due to limited sight in chain-of-thought reasoning. In contrast, the hierarchical structure of tree makes our method have broader sights and more error-tolerant; b) We bring uncertainty into reasoning and choose the most confident answer from *Closed-* and *Open-book QA*. However, IRCoT simply depends on the retrieved external knowledge, which may mislead the LLM reason-

ing; c) Our query tree represents the user intent very well. In contrast, IRCoT employs the last generated CoT sentence as the query, which only mentions the bridge entity, and is insufficient to represent the query intent accurately.

We also provide in-depth analyses of reasoning abilities in Table 1, from which we can observe that our method performs well on all types of questions, showing good generalizability and robustness. Compared with HotpotQA and 2WikiMQA, MuSiQue is more challenging and less cheatable, containing 2-4 hop questions with six different composition structures. Table 1 shows that the existing results on MuSiQue are low, especially for questions with complex compositional structure, *i.e.*, 3- or 4-hop questions. Our method improves the performance on these complex questions largely, with 11.8% F1 score on 3-hop questions. 2WikiMQA contains questions generated by hand-crafted templates. They contain the complex "Inference" questions that test the implicit reasoning ability, and "Bridge-comparison" questions that require both finding the bridge entities and doing comparisons to obtain the answer. Our methods improve largely on these questions, with 11.7% for "Inference" and 15.2% for "Bridge-comparison".

In addition, on HotpotQA, after adding the top-1 snippet from Google Search for each leaf question, ProbTree achieves a higher F1 result (64.1% v.s. 62.6%), demonstrating that better sub-modules can further improve the overall performance.

## 5.2 Ablation Study

We conduct a series of ablation studies to prove the effect of each component in ProbTree. The results are in Table 2.

### 5.2.1 Effect of Tree-of-thought Reasoning

We conduct two ablation studies: 1) **Sequential Decomposition (SD)**. It decomposes the complex question into a sequence of atomic questions, solves them step by step by selecting the most confident answer from *Close-book* and *Open-book QA* as in ProbTree, and takes the answer of the last atomic question as the final answer. For example, SD of $q^0$ in Fig. 2 is: $\langle q^4, q^5, q^2, q^3 \rangle^2$. Compared with ProbTree, F1 scores drop by 13.6%, 2.5%, and 2.6% on the three datasets, demonstrating the superiority of tree-based reasoning over chain-based

---

2Reference tokens in questions change accordingly. *E.g.*, in SD, $q^3$ is "When did #2 arrive in #3?"

| |
|---|
| **Question:** Which Canadian rock band released a song called "Counterparts" and had a drummer who was inducted into the Modern Drummer Hall of Fame? **Gold Answer:** Rush |

| |
|---|
| **Sequential Decomposition** |
| **Step 1:** Which Canadian rock band released a song called "Counterparts"? |
| **Answer 1:** The Canadian rock band Rush released a song called "Counterparts". So the answer is: Rush. |
| **Step 2:** Which drummer of #1 was inducted into the Modern Drummer Hall of Fame? *(incorrect sub-question)* |
| **Answer 2:** The drummer of Rush is Neil Peart. Neil Peart was inducted into the Modern Drummer Hall of Fame in 1983. So the answer is: Neil Peart. |
| **Final Answer:** Neil Peart |

| |
|---|
| **ProbTree (*Child-Aggregation QA*)** |
| **Context:** |
| Which Canadian rock band released a song called "Counterparts"? Rush |
| Which drummer of Rush was inducted into the Modern Drummer Hall of Fame? Neil Peart. *(incorrect sub-question)* |
| **Question:** |
| Which Canadian rock band released a song called "Counterparts" and had a drummer who was inducted into the Modern Drummer Hall of Fame? |
| **Answer:** The Canadian rock band Rush released a song called "Counterparts" and had a drummer, Neil Peart, who was inducted into the Modern Drummer Hall of Fame. So the answer is: Rush. |

Table 3: When generating unneccessary or incorrect sub-questions, SD cannot recover from the wrong decompositions, while the *Child-aggretating QA* in ProbTree enables meta-reasoning with the information from sub-questions to get the correct answer.

reasoning. Table 3 shows a case where ProbTree can recover from local errors, while SD cannot.

2) **ProbTree without *Child-aggregating QA* (w/o CA)**, where the answer of each node is selected from *Closed-* and *Open-book QA*. The performance drops on all the datasets. This is because *Closed-book QA* for complex questions often fails due to hallucination, and *Open-book QA* for complex questions also makes mistakes due to large number of distractor paragraphs (As shown in Table 11 in the Appendix). In contrast, *Child-aggregating QA* concisely summarizes information from sub-questions.

### 5.2.2 Effect of Probabilistic Reasoning

We bring uncertainty into reasoning for integrating parametric and external knowledge in a unifided famework. In this section, we prove the rationality of our confidence calculation method and the effect of knowledge integration.

| Dataset | Ev. | An. | Rt. | Rs. | De. | Cf. |
|---|---|---|---|---|---|---|
| HotpotQA | 38% | 16% | 8% | 6% | 16% | 16% |
| MuSiQue | 46% | 16% | 12% | 8% | 12% | 6% |
| 2WikiMQA | 46% | 8% | 14% | 14% | 8% | 10% |

Table 4: Errors of ProbTree can be grouped into Evaluation limitation (Ev.), Inaccurate Data Annotation (An.), Retrieval Error (Rt.), Reasoning Error (Rs.), Decomposition Error (De.) and Inaccurate Confidence (Cf.).

**Confidence:** To demonstrate the rationality of confidence calculation, we compare our method with (1) Randomly-choosing (RC) one of these three answers as the final answer; (2) Self-consistency (Wang et al., 2022b) that samples 3 (SC@3) or 5 (SC@5) CoTs for *Child-aggregating*, *Open-book* and *Close-book QA* respectively, and selects the most frequent answer from the 9/15 CoTs as the optimal answer (Details in Appendix C.3). Results in Table 2 show that ProbTree significantly outperforms RC, and beats SC@3 and SC@5 while using much fewer OpenAI API calls, demonstrating the effect of our confidence calculation method that takes the explanation likelihood as the indicator of answer confidence.

**Knowledge Intergration:** We respectively remove *Closed-* (w/o CB) and *Open-book QA* (w/o OB) from the overall framework and evaluate the final performances. As shown in Table 2, the F1 results drop for both cases. Fig. 2 shows an example from MuSiQue. For the sub-question *"Who founded Harvard Colledge"*, the *Open-book QA* is misled by a retrieved distractor sentence and gives a wrong prediction *"John Harvard"*, while the *Closed-book QA* gives the correct prediction *"Massachusetts General Court"* with higher confidence. On the other hand, for the sub-question *"When did Puritan arrive in New England?"*, the *Closed-book QA* hallucinates and gives a wrong prediction *"1620"*, while the *Open-book QA* gives the correct prediction *"1630"* with higher confidence based on the retrieved supporting paragraphs. From the case we can see probabilistic reasoning enables ProbTree to make better use of both parametric and external knowledge.

## 6 Error Analysis

We manually analyze 150 errors (50 per dataset) output by ProbTree. As shown in Table 4, the main reasons for error can be grouped into the following 6 categories: (1) *Evaluation Limitation*. Ac-

tually, the predictions are correct, however, they are measured as incorrect because our predictions are aliases of the gold answers, or questions with 1-to-N relations have multiple answers, while the gold annotations do not cover all of them. We attribute this type of error to the evaluation rather than the failure of our approach. (2) *Inaccurate Data Annotation*. We attribute this type of error to the dataset rather than the failure of our approach. For example, the annotated answer of the question in the released dataset is incorrect. (3) *Retrieval Error*. None of the retrieved paragraphs is relevant to the target question, therefore the *Open-book QA* does not work. (4) *Reasoning Error*. Supporting paragraph is retrieved but *Open-book QA* still gives a wrong prediction, or sub-questions have been correctly answered but *Child-aggregating QA* makes mistakes. (5) *Decomposition Error*. The generated query tree is incorrect, causing errors of the final answer. (6) *Inaccurate Confidence*: The correct prediction from *Open-book*, *Closed-book*, or *Child-aggregating QA* is not the most confident.

In all, Decomposition Error is most challenging in HotpotQA since it contains questions with complicated syntactic structures. Retrieval, reasoning, and decomposition errors make up more than 30% on each dataset, which may be reduced by using a more powerful retriever and backend LLM. Only a small percentage of errors are caused by the confidence mechanism of ProbTree.

## 7 Discussion and Conclusion

In this paper, we explore the LLM capacity of answering knowledge-intensive complex questions and propose a probabilistic tree-of-thought reasoning approach. Actually, we think this paper shows an example that LLMs leverage the strengths of specialized tools to accomplish complex tasks, *i.e.*, tool learning (Qin et al., 2023) in which LLMs start from understanding the user instruction, learn to decompose a complex task into several subtasks, and conquer each sub-task by selecting appropriate tools. In our case, LLMs start from understanding the complex question, decompose it into a query tree, and reason over the tree to solve questions by selecting an appropriate API from *Child-aggregating QA*, *Open-book QA*, and *Closed-book QA*. In the future, we hope to extend our approach to other complex tasks.

In addition, for Complex QA, in this paper, the external knowledge is limited to document corpora

which mainly contain unstructured text. We hope to cover other structured knowledge sources such as knowledge bases (KBs) (Wang et al., 2022a) and tables in the future, since knowledge from different sources complement each other. This can be achieved by incorporating other QA modules during reasoning such as semantic parsers that can execute on KBs or tables.

## 8 Limitations

ProbTree relies on a backend language model that (1) has good few-shot CoT reasoning ability; (2) is well-calibrated; (3) has a long context size. Currently, only several LLMs with over 100B parameters meet these requirements, which limits Prob-Tree to some extent.

For each node, ProbTree selects a most confident answer from *Open-book*, *Closed-book* and *Child-aggregating QA*, which brings additional computational costs compared to methods (Trivedi et al., 2022a; Press et al., 2022) that simply depend on retrieved knowledge. A possible improvement method is to incorporate active retrieval (Jiang et al., 2023) into the framework. In addition, as for external knowledge sources, we are limited to document corpora that mainly contain unstructured text. In the future, we can take other knowledge sources into consideration, such as structured knowledge bases and tables.

## 9 Ethical Considerations

LLMs suffer from hallucination, *i.e.*, generating factually incorrect information and are also possible to generate biased or offensive replies. Though retrieval-augmented approaches are expected to alleviate these issues, there is still a risk of being hacked through targeted means such as injecting misleading context into prompts or adding harmful knowledge into the retrieval corpora. Hence additional care and protective measures should be taken if such systems are deployed in user-facing applications.

All the datasets and encyclopedias used in this work are publicly published with permissible licenses.

## 10 Acknowledgements

This work is supported by grants from Cloud BU, Huawei Technologies, Tsinghua University Initiative Scientific Research Program, and the Institute for Guo Qiang, Tsinghua University (2019GQB0003).

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

## A Pilot Study for Our Confidence Mechanism

Estimating the confidence levels associated with the responses of LLMs is an important research topic. In the existing literature, methods can be grouped into three categories: 1) logits-based methods that focus on the probabilities derived from model logits; 2) verbalized confidence that directly asks a model to output its confidence; 3) consistency-based methods that yield multiple responses and use consistency as a surrogate for confidence.

Before initiating the reasoning framework, we first conducted a pilot study to investigate whether our explanation logits-based confidence is effective. As shown in Table 5, we compare different confidence calibration methods. For each method, given a question, we query LLMs multiple times (decode one response with greedy decoding, and sample another four with temperature t = 0.7), calculate the confidence score for each response, and take the most confident one as the final answer. Better answer F1 indicates better confidence calibration. We randomly sampled 500 questions from HotpotQA, and 500 from MuSiQue to conduct the pilot study. Answer F1 results show that our explanation logits-based confidence outperforms others, including the model itself estimating its uncertainty in language, and taking answer tokens as part of scoring.

In addition, we found that the average confidence score for all the 500 examples in HotpotQA is -0.142, and the average confidence score for examples with answer F1 1.0 is -0.109, and that of examples with answer F1 0.0 is -0.176. The low confidence of incorrect answers and high score of correct ones indicates the effect of our method.

## B Statistics of Test Sets

We use test sets provided by IRCoT (Trivedi et al., 2022a) for HotpotQA, MuSiQue, and 2WikiMQA. Specifically, they randomly sampled 500 questions from the original dev set of each dataset as the test set. The numbers of different types of questions in the test set of each dataset are listed in Table 9.

## C Implementation Details of Baselines

### C.1 Implementation Details of IRCoT

IRCoT (Trivedi et al., 2022a) uses code-davinci-002 as the backend LLM in the original paper, and we re-implement it with text-davinci-003 for a fair comparison. We use the same format of prompt as the original paper, and select five paragraphs for each demonstration example in the prompt, including all the supporting paragraphs and several distractor paragraphs. The key hyperparameter of IRCoT is $K$, the number of paragraphs to retrieve at each step. We select optimal $K \in \{2, 4, 6, 8\}$ based on the performance on the dev set.

Table 6 shows that the answer F1 results re-implemented by us are slightly lower than that of the original paper. We think the gap is due to two reasons: (1) the context size of text-davinci-003 is only half of that of code-davinci-002 (4097 v.s. 8192), so we can only use at most 5 demonstration examples in the prompt, while the original paper uses 15. (2) As shown in Table 7, the BM25-based retriever used in the original paper has a higher recall of supporting paragraphs than our implementation on MuSiQue and 2WikiMQA. Here the recall is computed according to 15 retrieved paragraphs using the original question as the query.

### C.2 Implementation Details of Self-Ask

Self-Ask (Press et al., 2022) uses text-davinci-002 and Google Search in the original paper, and we re-implement it with text-davinci-003 and BM25-based retriever. We follow Yoran et al. (2023) to prepend newly retrieved paragraphs to the original question. We use the same format of prompt as Yoran et al. (2023), and select five paragraphs for each demonstration example in the prompt, including all the supporting paragraphs and several distractor paragraphs. We retrieved top-$K$ paragraphs for each sub-question, where $K \in \{3, 5, 7\}$ is selected based on the dev set.

### C.3 Implementation Details of ProbTree with Self-consisitency

Suppose we respectively sample $n$ CoTs in *Closed-book*, *Open-book*, and *Child-aggregating QA* for each node in the query tree. We set the temperature $t = 0$ for the first CoT and $t = 0.7$ for the other $n-1$ CoTs. Then we collect answers after *"So the answer is:"* from all these $3n$ CoTs, and select the most frequent one as the optimal answer. If there are multiple most frequent answers with the same number of occurrences, we randomly select one.

| Datasets | logits (explanation) | logits (explanation + ans) | logits (ans) | verbalize | consistency | vanilla |
|---|---|---|---|---|---|---|
| HotpotQA | 47.8 | 47.5 | 43.7 | 44.5 | 47.6 | 47.1 |
| MuSiQue | 26.0 | 25.4 | 24.1 | 22.0 | 25.7 | 25.2 |

Table 5: Pilot study to investigate whether our explanation-based confidence is effective. Logits (explanation + ans) means answer tokens themselves are part of the scoring.

| Method | HotpotQA | MuSiQue | 2WikiMQA |
|---|---|---|---|
| IRCoT (ours) | 60.2 | 34.2 | 63.8 |
| IRCoT (Trivedi et al., 2022a) | 61.2 | 35.5 | 65.2 |
| ProbTree | **64.1** | **41.5** | **71.8** |

Table 6: Answer F1 results of IRCoT re-implemented by us, IRCoT in the original paper, and ProbTree.

| | HotpotQA | MuSiQue | 2WikiMQA |
|---|---|---|---|
| IRCoT (ours) | **64.2** | 37.5 | 53.7 |
| IRCoT (Trivedi et al., 2022a) | 61.5 | **44.6** | **68.1** |

Table 7: Recall of BM25-based retriever for IRCoT implemented by us and the original paper.

| Method | HotpotQA | MuSiQue | 2WikiMQA |
|---|---|---|---|
| ProbTree | **64.1** | **41.5** | **71.8** |
| w/o $ds^i$ | 63.9 | 41.3 | 71.6 |
| w/o descendants | 62.3 | 39.7 | **71.8** |

Table 8: Answer F1 results of ProbTree without decomposition score (w/o $ds^i$) or retrieved paragraphs from descendants (w/o descendants).

## D Additional Ablation Study

### D.1 Effect of Decomposing Score

To show the effect of considering question decomposing certainty in reasoning, we remove $ds^i$ from Eq. 9 and calculate $s_{ca}^i$ as:

$$s_{ca}^i = \frac{1}{|n|+1}\left( \sum_{j=1}^{|child^i|} s^{child_j^i} + \tilde{s}_{ca}^i \right). \qquad (11)$$

Table 8 shows that ignoring decomposing certainty causes a slight performance drop on all the datasets.

### D.2 Effect of Incorporating Retrieved Paragraphs from Descendants

In the *Open-book QA* module, the external knowledge for a non-leaf node $q^i$ contains the retrieved paragraphs for itself and all its descendants (Eq. 10). As shown in Table 8, removing retrieved paragraphs for descendants from $q^i.para$ causes a performance drop on both HotpotQA and MuSiQue.

## E Details of Other Recent Approaches

We introduce the details of other recent approaches. Their results are not generally apples-to-apples, and we report them as qualitative reference points. **ReAct** (Yao et al., 2022) interleaves reasoning, searching for information (action), and collecting retrieved knowledge (observation), until reaching the action of finalizing an answer. **ITER-RETGEN** (Shao et al., 2023). ITER-RETGEN leverages the CoT output from the previous iteration as the query to help retrieve more relevant knowledge, and outputs a new CoT output in the next iteration. **MCR** (Yoran et al., 2023). MCR meta-reasons over multiple chains of thought, mixes information between them, and selects the most relevant facts in generating an explanation and predicting the answer. We use the results of ReAct and ITER-RETGEN implemented by Shao et al. (2023). They use `text-davinci-003` as the backend LLM, choose the first 500 questions from the development sets of each dataset for evaluation, and use October 2017, December 2021, and December 2018 Wikipedia dump for HotpotQA, MuSiQue, and 2WikiMQA, respectively. They also fine-tune a dense retriever via distillation. MCR (Yoran et al., 2023) uses `code-davinci-002`, randomly samples 500 questions from the development set of each dataset for evaluation, and uses Google Search via SerpAPI as the retriever.

## F Percentage of Different QA Strategies

For the percentage of sub-questions that were answered using *Closed-book*, *Open-book*, and *Child-aggregating QA* respectively, we differentiate leaf and non-leaf questions because leaf nodes do not use *Child-aggregating QA*. As shown in Table 10, for non-leaf nodes, only less than 10% are answered using *Closed-book QA*, a small part of them are answered using *Open-book QA*, and most of them are answered using *Child-aggregating QA*, which shows that LLMs rely more on question decomposition when facing complex questions. For leaf nodes, LLMs rely on both internal and external knowledge, and rely more on external knowledge.

| | HotpotQA | | | MuSiQue | | | 2WikiMQA | | | |
|---|---|---|---|---|---|---|---|---|---|---|---|
| Overall | Bridge | Comparison | Overall | 2hop | 3hop | 4hop | Overall | Bridge | Inference | Comparison | Bridge-Comparison |
| 500 | 412 | 88 | 500 | 254 | 154 | 92 | 500 | 197 | 79 | 119 | 105 |

Table 9: Number of different types of questions in the test set of each dataset.

| Dataset | Leaf Nodes | | Non-leaf Nodes | | |
|---|---|---|---|---|---|
| | CB | OB | CB | OB | CA |
| HotpotQA | 23.0% | 77.0% | 8.1% | 31.7% | 60.2% |
| MuSiQue | 41.1% | 58.9% | 9.8% | 18.8% | 71.4% |
| 2WikiMQA | 17.7% | 82.3% | 1.3% | 11.7% | 87.0% |

Table 10: Percentage of different QA strategies.

# G    Prompts

To illustrate the prompt of ProbTree, we take 2WikiMQA as an example. We take the 20 manually written question-CoT pairs from IR-CoT (Trivedi et al., 2022a) as the base to create prompts for query tree generation and *Closed-book QA*. For query tree generation, we use the 20 questions from IRCoT, and manually write query trees for them (Fig. 5). For *Closed-book QA*, since the exemplars from IRCoT only contain complex questions, we add 4 exemplars for leaf nodes, and the final prompt contains 24 exemplars (Fig. 6). For *Open-book QA*, we mannualy write 3 annotations for leaf nodes (Fig. 7), and another 3 for non-leaf nodes (Fig. 8). For *Child-aggregating QA*, we mannually write 4 exemplars (Fig. 9). We also show the prompt (Fig. 10) for SD that is designed as an ablation study. Specifically, we take the 20 questions from IRCoT and manually write sequential decompositions for them.

# H    Case Study

We show a case in Table 11, in which the *Child-aggregating QA* gives the right answer by summarizing information from sub-questions, while *Open-book QA* makes a mistake due to large number of distractor paragraphs.

**Question:** Who is Philip III Of Navarre's father-in-law?
**Gold Answer:** Louis X of France

*Open-book QA*

**Prompt:**
#4 Wikipedia Title: Philip III of Navarre
Text: . . . he and his wife and cousin, Joan II of Navarre . . .
#6 Wikipedia Title: Blanche of Navarre, Queen of France
Text: Blanche of Navarre (1330 - 1398) was Queen of France as the wife of King Philip VI . . .
#8 Wikipedia Title: Joan II of Navarre
Text: . . . she was the only surviving child of Louis X of France.
Question:
Who is Philip III Of Navarre's father-in-law?
**Answer:** Philip III of Navarre was married to Blanche of Navarre. Blanche of Navarre's father was King Charles III of Navarre. Thus, Philip III of Navarre's father-in-law is King Charles III of Navarre. So the answer is: King Charles III of Navarre. ($s_{ob}$: -0.101)

*Child-aggregating QA*

**Prompt:**
Context:
Who is Philip III Of Navarre married to? Joan II of Navarre
Who is the father of Joan II of Navarre? Louis X of France.
Question:
Who is Philip III Of Navarre's father-in-law?
**Answer:** Philip III Of Navarre is married to Joan II of Navarre. The father of Joan II of Navarre is Louis X of France. Thus, the father-in-law of Philip III Of Navarre is Louis X of France. So the answer is: Louis X of France. ($s_{ca}$: -0.033)

Table 11: An illustration of *Open-book and Child-aggregating QA* for the root node. We mark misleading retrieved paragraphs and corresponding wrong prediction as red. In this case, *Open-book QA* makes a mistake due to large numbers of distractor paragraphs. In contrast, *Child-aggregating QA* concisely summarizes information from sub-questions and gives the right answer confidently.

Please generate a hierarchical question decomposition tree (HQDT) with json format for a given question. In this tree, the root node is the original complex question, and each non-root node is a sub-question of its parent. The leaf nodes are atomic questions that cannot be further decomposed.

Q: When did the director of film Hypocrite (Film) die?
A: {"When did the director of film Hypocrite (Film) die?": ["Who is the director of film Hypocrite (Film)?", "When did #1 die?"]}.
Q: Do director of film Coolie No. 1 (1995 Film) and director of film The Sensational Trial have the same nationality?
A: {"Do director of film Coolie No. 1 (1995 Film) and director of film The Sensational Trial have the same nationality?": ["What is the nationality of the director of film Coolie No. 1 (1995 Film)?", "What is the nationality of the director of film The Sensational Trial?"], "What is the nationality of the director of film Coolie No. 1 (1995 Film)?": ["Who is the director of film Coolie No. 1 (1995 Film)?", "What is the nationality of #1?"], "What is the nationality of the director of film The Sensational Trial?": ["Who is the director of film The Sensational Trial?", "What is the nationality of #1?"]}.
Q: Are both Kurram Garhi and Trojkrsti located in the same country?
A: {"Are both Kurram Garhi and Trojkrsti located in the same country?": ["Which country is Kurram Garhi located in?", "Which country is Trojkrsti located in?"]}.
Q: Who was born first out of Martin Hodge and Ivania Martinich?
A: {"Who was born first out of Martin Hodge and Ivania Martinich?": ["When was Martin Hodge born?", "When was Ivania Martinich born?"]}.
Q: Which film came out first, The Night Of Tricks or The Genealogy?
A: {"Which film came out first, The Night Of Tricks or The Genealogy?": ["When was the film The Night Of Tricks published?", "When was the film The Genealogy published?"]}.
Q: When did the director of film Laughter In Hell die?
A: {"When did the director of film Laughter In Hell die?": ["Who is the director of film Laughter In Hell?", "When did #1 die?"]}.
Q: Which film has the director died later, The Gal Who Took the West or Twenty Plus Two?
A: {"Which film has the director died later, The Gal Who Took the West or Twenty Plus Two?": ["When did the director of film The Gal Who Took the West die?", "When did the director of film Twenty Plus Two die?"], "When did the director of film The Gal Who Took the West die?": ["Who is the director of film The Gal Who Took the West?", "When did #1 die?"], "When did the director of film Twenty Plus Two die?": ["Who is the director of film Twenty Plus Two?", "When did #1 die?"]}.
Q: Who is Boraqchin (Wife Of ÃUgedei)'s father-in-law?
A: {"Who is Boraqchin (Wife Of ÃUgedei)'s father-in-law?": ["Who is Boraqchin married to?", "Who is the father of #1?"]}.
Q: What is the cause of death of Grand Duke Alexei Alexandrovich Of Russia's mother?
A: {"What is the cause of death of Grand Duke Alexei Alexandrovich Of Russia's mother?": ["Who is the mother of Grand Duke Alexei Alexandrovich Of Russia?", "What is the cause of death of #1?"]}.
Q: Which film has the director died earlier, When The Mad Aunts Arrive or The Miracle Worker (1962 Film)?
A: {"Which film has the director died earlier, When The Mad Aunts Arrive or The Miracle Worker (1962 Film)?": ["When did the director of film When The Mad Aunts Arrive die?", "When did the director of film The Miracle Worker (1962 Film) die?"], "When did the director of film When The Mad Aunts Arrive die?": ["Who is the director of film When The Mad Aunts Arrive?", "When did #1 die?"], "When did the director of film The Miracle Worker (1962 Film) die?": ["Who is the director of film The Miracle Worker (1962 Film)?", "When did #1 die?"]}.
Q: Which album was released earlier, What'S Inside or Cassandra'S Dream (Album)?
A: {"Which album was released earlier, What'S Inside or Cassandra'S Dream (Album)?": ["When was the album What'S Inside released?", "When was the album Cassandra'S Dream (Album) released?"]}.
Q: Are both mountains, Serre Mourene and Monte Galbiga, located in the same country?
A: {"Are both mountains, Serre Mourene and Monte Galbiga, located in the same country?": ["Which country was the mountain Serre Mourene located in?", "Which country was the mountain Monte Galbiga located in?"]}.
Q: What is the date of birth of the director of film Best Friends (1982 Film)?
A: {"What is the date of birth of the director of film Best Friends (1982 Film)?": ["Who is the director of film Best Friends (1982 Film)?", "What is the date of birth of #1?"]}.
Q: Which film has the director born first, Two Weeks With Pay or Chhailla Babu?
A: {"Which film has the director born first, Two Weeks With Pay or Chhailla Babu?": ["When was the director of film Two Weeks With Pay born?", "When was the director of film Chhailla Babu born?"], "When was the director of film Two Weeks With Pay born?": ["Who is the director of film Two Weeks With Pay?", "When was #1 born?"], "When was the director of film Chhailla Babu born?": ["Who is the director of film Chhailla Babu?", "When was #1 born?"]}.
Q: Who is the grandchild of Krishna Shah (Nepalese Royal)?
A: {"Who is the grandchild of Krishna Shah (Nepalese Royal)?": ["Who is the child of Krishna Shah (Nepalese Royal)?", "Who is the child of #1?"]}.
Q: When was the director of film P.S. Jerusalem born?
A: {"When was the director of film P.S. Jerusalem born?": ["Who is the director of film P.S. Jerusalem?", "When was #1 born?"]}.
Q: Which album was released more recently, If I Have to Stand Alone or Answering Machine Music?
A: {"Which album was released more recently, If I Have to Stand Alone or Answering Machine Music?": ["When was the album If I Have to Stand Alone released?", "When was the album Answering Machine Music released?"]}.
Q: Where did the director of film Maddalena (1954 Film) die?
A: {"Where did the director of film Maddalena (1954 Film) die?": ["Who is the director of film Maddalena (1954 Film)?", "Where did #1 die?"]}.
Q: When did the director of film The Boy And The Fog die?
A: {"When did the director of film The Boy And The Fog die?": ["Who is the director of film The Boy And The Fog?", "When did #1 die?"]}.
Q: Are the directors of films The Sun of the Sleepless and Nevada (1927 film) both from the same country?
A: {"Are the directors of films The Sun of the Sleepless and Nevada (1927 film) both from the same country?": ["Which country is the director of film The Sun of the Sleepless from?", "Which country is the director of film Nevada (1927 film) from?"], "Which country is the director of film The Sun of the Sleepless from?": ["Who is the director of film The Sun of the Sleepless?", "Which country is #1 from?"], "Which country is the director of film Nevada (1927 film) from?": ["Who is the director of film Nevada (1927 film)?", "Which country is #1 from?"]}.

Figure 5: Instruction and exemplars for the 2WikiMQA query tree generation prompt.

**Please answer the question by thinking step-by-step.**

Q: When did the director of film Hypocrite (Film) die?
A: The film Hypocrite was directed by Miguel Morayta. Miguel Morayta died on 19 June 2013. So the answer is: 19 June 2013.
Q: Do director of film Coolie No. 1 (1995 Film) and director of film The Sensational Trial have the same nationality?
A: Coolie No. 1 (1995 film) was directed by David Dhawan. The Sensational Trial was directed by Karl Freund. David Dhawan's nationality is India. Karl Freund's nationality is Germany. Thus, they do not have the same nationality. So the answer is: no.
Q: Are both Kurram Garhi and Trojkrsti located in the same country?
A: Kurram Garhi is located in the country of Pakistan. Trojkrsti is located in the country of Republic of Macedonia. Thus, they are not in the same country. So the answer is: no.
Q: Who was born first out of Martin Hodge and Ivania Martinich?
A: Martin Hodge was born on 4 February 1959. Ivania Martinich was born on 25 July 1995. Thus, Martin Hodge was born first. So the answer is: Martin Hodge.
Q: Which film came out first, The Night Of Tricks or The Genealogy?
A: The Night of Tricks was published in the year 1939. The Genealogy was published in the year 1979. Thus, The Night of Tricks came out first. So the answer is: The Night Of Tricks.
Q: When did the director of film Laughter In Hell die?
A: The film Laughter In Hell was directed by Edward L. Cahn. Edward L. Cahn died on August 25, 1963. So the answer is: August 25, 1963.
Q: Which film has the director died later, The Gal Who Took the West or Twenty Plus Two?
A: The film Twenty Plus Two was directed by Joseph M. Newman. The Gal Who Took the West was directed by Frederick de Cordova. Joseph M. Newman died on January 23, 2006. Fred de Cordova died on September 15, 2001. Thus, the person to die later from the two is Twenty Plus Two. So the answer is: Twenty Plus Two.
Q: Who is Boraqchin (Wife Of ÃUgedei)'s father-in-law?
A: Boraqchin is married to ÃUgedei Khan. ÃUgedei Khan's father is Genghis Khan. Thus, Boraqchin's father-in-law is Genghis Khan. So the answer is: Genghis Khan. A: The mother of Grand Duke Alexei Alexandrovich of Russia is Maria Alexandrovna. Maria Alexandrovna died from tuberculosis. So the answer is: tuberculosis.
Q: Which film has the director died earlier, When The Mad Aunts Arrive or The Miracle Worker (1962 Film)?
A: When The Mad Aunts Arrive was directed by Franz Josef Gottlieb. The Miracle Worker (1962 film) was directed by Arthur Penn. Franz Josef Gottlieb died on 23 July 2006. Arthur Penn died on September 28, 2010. Thus, of the two, the director to die earlier is Franz Josef Gottlieb, who directed When The Mad Aunts Arrive. So the answer is: When The Mad Aunts Arrive.
Q: Which album was released earlier, What'S Inside or Cassandra'S Dream (Album)?
A: What's Inside was released in the year 1995. Cassandra's Dream (album) was released in the year 2008. Thus, of the two, the album to release earlier is What's Inside. So the answer is: What's Inside.
Q: Are both mountains, Serre Mourene and Monte Galbiga, located in the same country?
A: Serre Mourene is located in Spain. Monte Galbiga is located in Italy. Thus, the two countries are not located in the same country. So the answer is: no.
Q: What is the date of birth of the director of film Best Friends (1982 Film)?
A: The film Best Friends was directed by Norman Jewison. Norman Jewison was born on July 21, 1926. So the answer is: July 21, 1926.
Q: Which film has the director born first, Two Weeks With Pay or Chhailla Babu?
A: Two Weeks with Pay was directed by Maurice Campbell. Chhailla Babu was directed by Joy Mukherjee. Maurice Campbell was born on November 28, 1919. Joy Mukherjee was born on 24 February 1939. Thus, from the two directors, Chhailla Babu was born first, who directed Two Weeks With Pay. So the answer is: Two Weeks With Pay.
Q: Who is the grandchild of Krishna Shah (Nepalese Royal)?
A: Krishna Shah has a child named Rudra Shah. Rudra Shah has a child named Prithvipati Shah. Thus, Krishna Shah has a grandchild named Prithvipati Shah. So the answer is: Prithvipati Shah.
Q: When was the director of film P.S. Jerusalem born?
A: P.S. Jerusalem was directed by Danae Elon. Danae Elon was born on December 23, 1970. So the answer is: December 23, 1970.
Q: Which album was released more recently, If I Have to Stand Alone or Answering Machine Music?
A: If I Have to Stand Alone was published in the year 1991. Answering Machine Music was released in the year 1999. Thus, of the two, the album to release more recently is Answering Machine Music. So the answer is: Answering Machine Music.
Q: Where did the director of film Maddalena (1954 Film) die?
A: The film Maddalena is directed by Augusto Genina. Augusto Genina died in Rome. So the answer is: Rome.
Q: When did the director of film The Boy And The Fog die?
A: The director of The Boy and the Fog is Roberto GavaldÃ¸sn. Roberto GavaldÃ¸sn died on September 4, 1986. So the answer is: September 4, 1986.
Q: Are the directors of films The Sun of the Sleepless and Nevada (1927 film) both from the same country?
A: The director of Sun of the Sleepless is Temur Babluani. The director of Nevada (1927 film) is John Waters. John Waters is from the country of America. Temur Babluani is from the country of Georgia. Thus, John Walters and Temur Babluani are not from the same country. So the answer is: no.
Q: Who is the director of film Hypocrite (Film)?
A: The film Hypocrite was directed by Miguel Morayta. So the answer is: Miguel Morayta.
Q: When did Franz Josef Gottlieb die?
A: Franz Josef Gottlieb died on 23 July 2006. So the answer is: 23 July 2006.
Q: When was the album What'S Inside released?
A: What's Inside was released in the year 1995. So the answer is: 1995.
Q: Which country was the mountain Serre Mourene located in?
A: Serre Mourene is located in Spain. So the answer is Spain.

Figure 6: Instruction and exemplars for the 2WikiMQA *Closed-book QA* prompt, including 24 exemplars.

**Given a question and the relevant Wikipedia text, answer the question and explain why. If you are unsure, answer Unknown.**

#1 Wikipedia Title: Hypocrite (film)
Text: Hypocrite (Spanish: Hipócrita..!) is a 1949 Mexican thriller film directed by Miguel Morayta and starring Antonio Badú, Leticia Palma, Carmen Molina and Luis Beristáin. The film included the song "Hipócrita". The film's sets were designed by Francisco Marco Chillet.
#2 Wikipedia Title: Who? (film)
Text: Who? is a 1974 film based on the 1958 novel of the same name by Algis Budrys. It was directed by Jack Gold and stars Elliott Gould, Trevor Howard, and Joseph Bova. Some video releases were retitled "The Man in the Steel Mask" or "Roboman".
#3 Wikipedia Title: Who Is the Guilty?
Text: Who is the Guilty? ( sometimes" Who is to Blame?") is a 1925 Georgian silent film directed by Alexandre Tsutsunava
#4 Wikipedia Title: Who Is the Man?
Text: Who Is The Man?( 1924) is a British silent film drama directed by Walter Summers. The film was based on the successful French play" Daniel" by Louis Verneuil and is notable as the first screen appearance of John Gielgud.
#5 Wikipedia Title: Deceit (1923 film)
Text: Deceit( sometimes referred to as The Deceit) is a 1923 American silent black- and- white film. It is a conventional melodrama directed by Oscar Micheaux. Like many of Micheauxś films," Deceit" casts clerics in a negative light. Although the film was shot in 1921, it was not released until 1923. It is not known whether the film currently survives, which suggests that it is a lost film. The 1922 film" The Hypocrite" was shown within" Deceit" as a film within a film.
Q: Who is the director of film Hypocrite (Film)?
A: The film Hypocrite is directed by Miguel Morayta. So the answer is: Miguel Morayta.

#1 Wikipedia Title: Kurram Garhi
Text: Kurram Garhi is a small village located near the city of Bannu, which is the part of Khyber Pakhtunkhwa province of Pakistan. Its population is approximately 35000. Barren hills are near this village. This village is on the border of Kurram Agency. Other nearby villages are Peppal, Surwangi and Amandi Kala.
#2 Wikipedia Title: Kurram Garhi Hydropower Plant
Text: Kurram Garhi Hydropower Plant( KGHPP) is a small, low- head, run- of- the- river hydroelectric power generation station of 4.0 megawatt generation capacity( four units of 1.0 MW each), located at Kurram Garhi, a small town in Bannu KPK province of Pakistan on the flows of Kuchkot Canal from Kurram River. It is a small hydel power generating plant constructed and put in commercial operation on February 1958 with the Average Annual generating capacity of 17 million units( GWh) of least expensive electricity.
#3 Wikipedia Title: Country Is
Text: " Country Is" is a song written and recorded by American country music artist Tom T. Hall. It was released in September 1974 as the second and final single from the album of the same name," Country Is". The song was Hallś fifth number one on the country chart. The single went to number one for a single week and spent a total of eleven weeks on the country chart.
#4 Wikipedia Title: Which Way Is Up?
Text: Which Way is Up? is a 1977 American comedy film starring Richard Pryor and directed by Michael Schultz. It is a remake of the 1972 Italian comedy film" The Seduction of Mimi". Richard Pryor plays three roles: an orange picker who has two women at the same time, the orange pickerś father, and a reverend who gets the orange pickerś wife pregnant.
#5 Wikipedia Title: In Country
Text: In Country is a 1989 American drama film produced and directed by Norman Jewison, starring Bruce Willis and Emily Lloyd. The screenplay by Frank Pierson and Cynthia Cidre was based on the novel by Bobbie Ann Mason. The original music score was composed by James Horner. Willis earned a best supporting actor Golden Globe nomination for his role.
Q: Which country is Kurram Garhi located in?
A: Kurram Garhi is located in the country of Pakistan. So the answer is: Pakistan.

#1 Wikipedia Title: Neer Shah
Text: Neer Bikram Shah, also known as Nir Shah, is a Nepalese movie actor, a poet, lyricist, movie director, and businessman. He is related to the Royal family of Nepal.
#2 Wikipedia Title: Gajraj Mishra
Text: TRajguru Gajraj Mishra also spelled Gajaraj Mishra was a Nepalese politician, ambassador, diplomat and a royal priest of Shah dynasty. He was always inclined to his disciple Prince Regent Bahadur Shah of Nepal. Gajraj Mishra was disfavoured by his disciple King Pratap Singh Shah due to his support to Prince Bahadur Shah. He was also disfavoured by Pratap Singh's son Rana Bahadur Shah.
#3 Wikipedia Title: Princess Helen Shah of Nepal
Text: Princess Helen Shah of Nepal( September 21, 1926 – September 12, 2008) was a member of the former Nepalese royal family. She was the wife of Prince Basundhara of Nepal, a son of King Tribhuvan of Nepal and his second wife, Queen Ishwari.
#4 Wikipedia Title: Krishna Shah (Nepalese royal)
Text: Krishna Shah (?–1661) was the king of the Gorkha Kingdom in the Indian subcontinent, present-day Nepal. He was the father of Rudra Shah.
#5 Wikipedia Title: Ajaya Pratap Shah
Text: Ajay Pratap Shah( died September 12,? in Lucknow, India) was a Nepalese politician, belonging to the Rastriya Prajatantra Party. In 1999 parliamentary election he was elected from the Kapilvastu- 4 constituency, with 14091 votes. After the royal coup d'état in February 2005, Shah went into exile in India. After his death, RPP nominated his son, Abhisek Pratap Shah, to take his parliamentary seat in January 2008.
Q: Who is the child of Krishna Shah (Nepalese Royal)?
A: Krishna Shah was the father of Rudra Shah. So the answer is: Rudra Shah.

Figure 7: Instruction and examplars for the 2WikiMQA *Open-book QA* (leaf node) prompt, including 3 examplars.

**Given a question and the relevant Wikipedia text, answer the question and explain why. If you are unsure, answer Unknown.**

#1 Wikipedia Title: Hypocrite (film)
Text: Hypocrite (Spanish: Hipócrita..!) is a 1949 Mexican thriller film directed by Miguel Morayta and starring Antonio Badú, Leticia Palma, Carmen Molina and Luis Beristáin. The film included the song "Hipócrita". The film's sets were designed by Francisco Marco Chillet.
#2 Wikipedia Title: When the Legends Die
Text: When The Legends Die is a 1963 novel, by Hal Borland, and a DeLuxe Color film released in 1972 by Twentieth Century-Fox.
#3 Wikipedia Title: Who Is the Guilty?
Text: Who is the Guilty? ( sometimes" Who is to Blame?") is a 1925 Georgian silent film directed by Alexandre Tsutsunava
#4 Wikipedia Title: Miguel Morayta
Text: Miguel Morayta( 15 August 1907 – 19 June 2013) was a Spanish film director and screenwriter. He directed 74 films between 1944 and 1978. At the outbreak of the Spanish Civil War, Morayta was a Spanish artillery officer, who joined the Republican side. After Francisco Franco's victory, he left Spain for France and Africa, finally arriving in Mexico in 1941, where he started his career. He was living in Mexico when he died aged 105.
#5 Wikipedia Title: Joselito vagabundo
Text: Joselito vagabundo(" Joselito Vagabond") is a 1966 Mexican film. It stars Sara García and is directed by Miguel Morayta.
Q: When did the director of film Hypocrite (Film) die?
A: The film Hypocrite was directed by Miguel Morayta. Miguel Morayta died on 19 June 2013. So the answer is: 19 June 2013.

#1 Wikipedia Title: Kurram Garhi
Text: Kurram Garhi is a small village located near the city of Bannu, which is the part of Khyber Pakhtunkhwa province of Pakistan. Its population is approximately 35000. Barren hills are near this village. This village is on the border of Kurram Agency. Other nearby villages are Peppal, Surwangi and Amandi Kala.
#2 Wikipedia Title: Kurram Garhi Hydropower Plant
Text: Kurram Garhi Hydropower Plant( KGHPP) is a small, low- head, run- of- the- river hydroelectric power generation station of 4.0 megawatt generation capacity( four units of 1.0 MW each), located at Kurram Garhi, a small town in Bannu KPK province of Pakistan on the flows of Kuchkot Canal from Kurram River. It is a small hydel power generating plant constructed and put in commercial operation on February 1958 with the Average Annual generating capacity of 17 million units( GWh) of least expensive electricity.
#3 Wikipedia Title: Trojkrsti
Text: Trojkrsti is a village in Municipality of Prilep, Republic of Macedonia.
#4 Wikipedia Title: All Men Are the Same
Text: All Men Are the Same is a 1994 Spanish comedy film directed by Manuel Gómez Pereira.
#5 Wikipedia Title: The Both
Text: The Both is an American musical duo consisting of Aimee Mann and Ted Leo, both of whom had longstanding musical careers before beginning a collaboration in 2013. Their first album, self- titled" The Both", was released in April 2014.
Q: Are both Kurram Garhi and Trojkrsti located in the same country?
A: Kurram Garhi is located in the country of Pakistan. Trojkrsti is located in the country of Republic of Macedonia. Thus, they are not in the same country. So the answer is: no.

#1 Wikipedia Title: Krishna Shah (Nepalese royal)
Text: Krishna Shah (?–1661) was the king of the Gorkha Kingdom in the Indian subcontinent, present-day Nepal. He was the father of Rudra Shah.
#2 Wikipedia Title: Neer Shah
Text: Neer Bikram Shah, also known as Nir Shah, is a Nepalese movie actor, a poet, lyricist, movie director, and businessman. He is related to the Royal family of Nepal.
#3 Wikipedia Title: Gajraj Mishra
Text: Rajguru Gajraj Mishra also spelled Gajaraj Mishra was a Nepalese politician, ambassador, diplomat and a royal priest of Shah dynasty. He was always inclined to his disciple Prince Regent Bahadur Shah of Nepal. Gajraj Mishra was disfavoured by his disciple King Pratap Singh Shah due to his support to Prince Bahadur Shah. He was also disfavoured by Pratap Singh's son Rana Bahadur Shah.
#4 Wikipedia Title: Princess Helen Shah of Nepal
Text: Princess Helen Shah of Nepal( September 21, 1926 – September 12, 2008) was a member of the former Nepalese royal family. She was the wife of Prince Basundhara of Nepal, a son of King Tribhuvan of Nepal and his second wife, Queen Ishwari.
#5 Wikipedia Title: Rudra Shah
Text: Rudra Shah (?–1673) was the king of the Gorkha Kingdom in the Indian subcontinent, present-day Nepal. He was the father of Prithvipati Shah.
Q: Who is the grandchild of Krishna Shah (Nepalese Royal)?
A: Krishna Shah has a child named Rudra Shah. Rudra Shah has a child named Prithvipati Shah. Thus, Krishna Shah has a grandchild named Prithvipati Shah. So the answer is: Prithvipati Shah.

Figure 8: Instruction and examplars for the 2WikiMQA *Open-book QA* (non-leaf node) prompt, including 3 examplars.

**Given a question and a context, answer the question and explain why.**

#
Context:
Who is the director of film Hypocrite? Miguel Morayta.
When did Miguel Morayta die? 19 June 2013.

Question:
When did the director of film Hypocrite (Film) die?

Answer:
The film Hypocrite was directed by Miguel Morayta. Miguel Morayta died on 19 June 2013. So the answer is: 19 June 2013.
#
Context:
When was the director of film Two Weeks With Pay born? November 28, 1919.
When was the director of film Chhailla Babu born? 24 February 1939.

Question:
Which film has the director born first, Two Weeks With Pay or Chhailla Babu?

Answer:
The director of Two Weeks With Pay was born on November 28, 1919. The director of Chhailla Babu With Pay was born on 24 February 1939. Thus, Two Weeks With Pay has the director born first. So the answer is: Two Weeks With Pay.
#
Context:
Where is Phu Luong located? Vietnam.
Which country is Vietnam in? Southeast Asia.

Question:
Which country contains Phu Luong?

Answer:
Phu Luong is located in the country Vietnam. So the answer is: Vietnam.
#
Context:
Who is the mother of Abraham Lincoln? Nancy Hanks Lincoln.
Who is the father of Nancy Hanks Lincoln? James Hanks.
Who is the father of James Hanks? Joseph Hanks.

Question:
Who is the maternal grandfather of Abraham Lincoln?

Answer:
The mother of Abraham Lincoln is Nancy Hanks Lincoln. The father of Nancy Hanks Lincoln is James Hanks. Thus, the maternal grandfather of Abraham Lincoln is James Hanks. So the answer is: James Hanks.
#

Figure 9: Instruction and examplars for the 2WikiMQA *Child-aggregating QA* prompt, including 4 examplars.

**Please decompose the question into sub-questions.**

Q: When did the director of film Hypocrite (Film) die?
A: ["Who is the director of film Hypocrite (Film)?", "When did #1 die?"]
Q: Do director of film Coolie No. 1 (1995 Film) and director of film The Sensational Trial have the same nationality?
A: ["Who is the director of film Coolie No. 1 (1995 Film)?", "What is the nationality of #1?, "Who is the director of film The Sensational Trial?", "What is the nationality of #3?", "Are #2 and #4 the same country?"]
Q: Are both Kurram Garhi and Trojkrsti located in the same country?
A: ["Which country is Kurram Garhi located in?", "Which country is Trojkrsti located in?", "Are #1 and #2 the same country?"]
Q: Who was born first out of Martin Hodge and Ivania Martinich?
A: ["When was Martin Hodge born?", "When was Ivania Martinich born?", "Martin Hodge was born on #1. Ivania Martinich was born on #2. Which of they was born first?"]
Q: Which film came out first, The Night Of Tricks or The Genealogy?
A: ["When was the film The Night Of Tricks published?", "When was the film The Genealogy published?", "The Night Of Tricks was published on #1. The Genealogy was published on #2. Which of them came out first?"]
Q: When did the director of film Laughter In Hell die?
A: ["Who is the director of film Laughter In Hell?", "When did #1 die?"]
Q: Which film has the director died later, The Gal Who Took the West or Twenty Plus Two?
A: ["Who is the director of film The Gal Who Took the West?", "When did #1 die?", "Who is the director of film Twenty Plus Two?", "When did #3 die?", "The director of The Gal Who Took the West died on #2. The director of Twenty Plus Two died on #4. Which of these two film has the director died later?"]
Q: Who is Boraqchin (Wife Of ÃUgedei)'s father-in-law?
A: ["Who is Boraqchin married to?", "Who is the father of #1?"]
Q: What is the cause of death of Grand Duke Alexei Alexandrovich Of Russia's mother?
A: ["Who is the mother of Grand Duke Alexei Alexandrovich Of Russia?", "What is the cause of death of #1?"]
Q: Which film has the director died earlier, When The Mad Aunts Arrive or The Miracle Worker (1962 Film)?
A: ["Who is the director of film When The Mad Aunts Arrive?", "When did #1 die?", "Who is the director of film The Miracle Worker (1962 Film)?", "When did #3 die?", "The director of When The Mad Aunts Arrive died on #2. The director of The Miracle Worker (1962 Film) died on #4. Which of these two film has the director died earlier?"]
Q: Which album was released earlier, What'S Inside or Cassandra'S Dream (Album)?
A: ["When was the album What'S Inside released?", "When was the album Cassandra'S Dream (Album) released?", "What'S Inside was released on #1. Cassandra'S Dream (Album) was released on #2. Which of them was released earlier?"]
Q: Are both mountains, Serre Mourene and Monte Galbiga, located in the same country?
A: ["Which country was the mountain Serre Mourene located in?", "Which country was the mountain Monte Galbiga located in?", "Are #1 and #2 the same country?"]
Q: What is the date of birth of the director of film Best Friends (1982 Film)?
A: ["Who is the director of film Best Friends (1982 Film)?", "What is the date of birth of #1?"]
Q: Which film has the director born first, Two Weeks With Pay or Chhailla Babu?
A: ["Who is the director of film Two Weeks With Pay?", "When was #1 born?", "Who is the director of film Chhailla Babu?", "When was #3 born?", "The director of Two Weeks With Pay was born on #2. The director of film Chhailla Babu was born on #4. Which of these two film has the director born first?"]
Q: Who is the grandchild of Krishna Shah (Nepalese Royal)?
A: ["Who is the child of Krishna Shah (Nepalese Royal)?", "Who is the child of #1?"]
Q: When was the director of film P.S. Jerusalem born?
A: ["Who is the director of film P.S. Jerusalem?", "When was #1 born?"]
Q: Which album was released more recently, If I Have to Stand Alone or Answering Machine Music?
A: ["When was the album If I Have to Stand Alone released?", "When was the album Answering Machine Music released?", "I Have to Stand Alone was released on #1. Answering Machine Music was released on #2. Which of them was released more recently?"]
Q: Where did the director of film Maddalena (1954 Film) die?
A: ["Who is the director of film Maddalena (1954 Film)?", "Where did #1 die?"]
Q: When did the director of film The Boy And The Fog die?
A: ["Who is the director of film The Boy And The Fog?", "When did #1 die?"]
Q: Are the directors of films The Sun of the Sleepless and Nevada (1927 film) both from the same country?
A: ["Who is the director of film The Sun of the Sleepless?", "Which country is #1 from?", "Who is the director of film Nevada (1927 film)?", "Which country is #3 from?", "Are #2 and #4 the same country?"]

Figure 10: Instruction and examplars for the 2WikiMQA sequential decomposition generation prompt, including 20 examplars.