# OpenReview forum: "Probabilistic Tree-of-thought Reasoning for Answering Knowledge-intensive Complex Questions"
_EMNLP/2023/Conference — EMNLP 2023 Findings_

### Official Review · Reviewer_ahcZ · 2023-08-03

**Soundness:** 3

**Excitement:**

4: Strong: This paper deepens the understanding of some phenomenon or lowers the barriers to an existing research direction.

**Paper Topic And Main Contributions:**

Recently, chain-of-thought (CoT) has been used to improve the results in tasks like question answering, trough converting a complex question into a number of simpler subquestions. This work argues two main limitations in this solution: (1) its sequential nature will result in incorrect final answers due to at least 1 incorrect answer in the chain of subquestions and (2) any mistake in the retrieval step will likely cause a mistake in the proceedings steps (misleading the reasoning).

This paper proposes replacing CoT reasoning with Tree-of-thought reasoning, where a complex question is decomposed into a tree. Every node in the tree represents a subquestion. The process of answering starts at the bottom with leaf nodes (questions that cannot be further decomposed). Going up, subquestions are answered over LLMs using zero-shot prompts, external-knowledge-augmented prompts, or an aggregation from previous answers. Experiments show that this new methodology outperforms previous CoT methods.

**Questions For The Authors:**

Q1: on the error analysis: did you find any errors caused by questions which included wrong assumptions, e.g., who was the first female president of the united states? Are there many of those and how would your model handle them?
Q2: Curious about the % of subquestions that were answered using CB, OB, and CA respectively.
Q3: Will you publish the code and data for reproducibility?

**Reasons To Accept:**

- The problem is well-motivated.
- The methodology is intuitive and well-explained.
- Experiments against SOTA methods seem sufficient.


**Reasons To Reject:**

- Lack of intrinsic experiment with various instantiations  of the ProbTree method's parameters, e.g., choice of LLM, choice of retrieval model, choice of external knowledge source.

**Reproducibility:**

4: Could mostly reproduce the results, but there may be some variation because of sample variance or minor variations in their interpretation of the protocol or method.

**Reviewer Confidence:**

3: Pretty sure, but there's a chance I missed something. Although I have a good feel for this area in general, I did not carefully check the paper's details, e.g., the math, experimental design, or novelty.

**Typos Grammar Style And Presentation Improvements:**

- Table 4 is not mentioned (referred to) in the text.
- Line 879 typo

---

> ### Author Rebuttal · Authors · 2023-08-28
>
> We are very pleased to know that the reviewer finds our work is well-motivated and intuitive with sufficient experimental results.
>
> ### Questions with Wrong Assumptions
> **Q**: Did you find any errors caused by questions that included wrong assumptions, e.g., who was the first female president of the united states? Are there many of those and how would your model handle them?
>
> **A**: Yes, our model can handle them, though the benchmark datasets themselves don't contain such questions. Our model can say "none" or "unknown" for questions with wrong assumptions. For the question *"who was the first female president of the united states?"*, our system chooses the most confident one from CB and OB shown below, so the final answer is "none".
>
> | QA Strategy         | Output | Confidence |
> |----------|:--------:|:--------:|
> | CB   |   The first female president of the United States has not yet been elected. So the answer is: none.  | -0.137 |
> | OB    |   Unknown. The Wikipedia article "Female president of the United States in popular culture" does not mention any real-life female presidents of the United States, only fictional ones. So the answer is: Unknown.| -0.269 |
>
> ### CA, CB, OB Percentage
> **Q**: Curious about the % of subquestions that were answered using CB, OB, and CA respectively.
>
> **A**: Thank you for your question. The results are as follows:
>
> |          | leaf questions |       | non-leaf questions |       |       |
> |----------|:--------------:|:-----:|:------------------:|:-----:|:-----:|
> |          |       CB       |   OB  |         CB         |   OB  |   CA  |
> | HotpotQA |      23.0%     | 77.0% |        8.1%        | 31.7% | 60.2% |
> | MuSiQue  |      41.1%     | 58.9% |        9.8%        | 18.8% | 71.4% |
> | 2WikiMQA |      17.7%     | 82.3% |        1.3%        | 11.7% | 87.0% |
>
> We differentiate leaf and non-leaf questions because leaf nodes do not use CA. For non-leaf nodes, only less than 10% are answered using CB, a small part of them are answered using OB, and most of them are answered using CA, which shows that LLMs rely more on question decomposition when facing complex questions that need multi-step reasoning. For leaf nodes, LLMs rely on both internal and external knowledge, and rely more on external knowledge. We will add this investigation to our revision.
>
> ### Publish Codes
> **Q**: Will you publish the code and data for reproducibility?
>
> **A**: Yes, we will publish the codes and data upon acceptance.
>
> ### More Intrinsic Experiments
> **Q**: Lack of intrinsic experiment with various instantiations of the ProbTree method's parameters, e.g., choice of LLM, choice of retrieval model, choice of external knowledge source.
>
> **A**: Thank you for your advice. The main reason we choose them is to follow the tradition of previous works for fair comparison. We will add more experiments with different models, retrivers and external knowledge sources in our revision.

---

### Official Review · Reviewer_M8kU · 2023-08-04

**Typos Grammar Style And Presentation Improvements:** N/A
**Soundness:** 3

**Excitement:**

2: Mediocre: This paper makes marginal contributions (vs non-contemporaneous work), so I would rather not see it in the conference.

**Missing References:**

N/A

**Paper Topic And Main Contributions:**

This paper proposes a novel approach to solving complex questions with probabilistic CoT reasoning over a query tree, eliminating the negative retrieval problem and limited sight of chain-based methods. The paper makes several contributions, including exploring the LLM's capacity to decompose complex questions and proposing the probabilistic tree-of-thought reasoning approach. It brings uncertainty into reasoning and proposes a self-evaluation-based method to quantify answer confidence. Then, different kinds of reasoning mechanisms can be integrated into a unified framework and automatically chosen to conduct reasoing. Last, it conducts comprehensive experiments and evaluates the effect of each component with careful ablation studies.

**Questions For The Authors:**

1. Please comprehensively explain the difference between the contributions in this paper and [1].
2. Can some simple semantic parsing method be used to decompose the question? Why is the maximum number of nodes in each layer set to 3? Can different numbers be used for decomposition?
3. Can uncertainty estimation of the intermediate answers be used with other reasoning frameworks, such as sequential decomposition?
4. Does the uncertainty truly reflect the quality of the decomposition and the accuracy of the answers? Case studies or statistics of the consistency between incorrect answers and uncertainty scores are desired.
5. Why child-aggregation reasoning can generate accurate answers with incorrect sub-question context shown in Table 3?

[1] Zhang, Jiajie, et al. "Reasoning over Hierarchical Question Decomposition Tree for Explainable Question Answering."  ACL 2023.

**Reasons To Accept:**

1. It presents a probabilistic tree-of-thought reasoning framework, which not only decomposes the reasoning process on complex questions into a tree structure but also considers the uncertainty of the intermediate answers to eliminate the error.
2. It presents a self-evaluation-based method to quantify answer confidence.
3. The experiment results are significant and promising. The overall presentation of the paper is good and easy to understand.

**Reasons To Reject:**

1. The idea of question decomposition, uncertainty estimation, and even the notations used in this paper are highly similar to reference [1], which diminishes the contributions and novelties of this paper.
2. The decomposition of the questions could be further discussed. Can we just use some simple semantic parsing method to decompose the question?
3. The calculated uncertainty could be further analyzed. Whether the uncertainty can truly reflect the quality of the decomposition and the accuracy of the answers is still unclear.
4. The process of extracting the intermediate answers for the next step of reasoning is not discussed.

[1] Zhang, Jiajie, et al. "Reasoning over Hierarchical Question Decomposition Tree for Explainable Question Answering."  ACL 2023.

**Reproducibility:**

4: Could mostly reproduce the results, but there may be some variation because of sample variance or minor variations in their interpretation of the protocol or method.

**Reviewer Confidence:**

4: Quite sure. I tried to check the important points carefully. It's unlikely, though conceivable, that I missed something that should affect my ratings.

---

> ### Author Rebuttal · Authors · 2023-08-28
>
> We are glad to hear that the reviewer finds our experiment results are significant and the presentation is easy to follow.
>
> ### Contribution
> **Q**: Please comprehensively explain the difference between the contributions in this paper and [1].
>
> **A**: We admit that reasoning on tree-like computation graphs has been proposed before. It's a research methodology whose core is to derive tree-like computation graphs and reason with uncertainty estimation. In this paper, we are exploring whether this methodology can help improve LLMs' ability, i.e., reduce hallucination in model generation, which is a crucial issue for today's LLMs.
>
> Specifically, we propose to integrate internal (parametric) and external (non-parametric) knowledge in a probabilistic way for LLMs. And during realization, due to the characteristics of LLMs, several unique challenges are posed: 1) how to estimate the answer confidence with the LLM itself is an open problem; 2) how to propagate confidence scores globally when there are local mistakes caused by inaccurate in-context learning such as wrong decompositions.
>
> For 1), given the observation that LLMs tend to be well-calibrated and low probability often indicates a lack of knowledge, we hypothesize that likelihood of the explanation (i.e., reasoning steps in CoT) indicates LLM confidence in the answer, and propose to quantify answer confidence with explanation logits, which is demonstrated effective with various experimental results. For 2), we propose Child-aggregating QA to globally reason with the information from child nodes which makes recovering from local mistakes possible, such as wrong question decompositions and incorrectly answered sub-questions. Reasoning with Child-aggregating QA, Open-book QA and Closed-book QA comprehensively reduces the hallucination and improves the reasoning abilities of LLMs significantly.
>
> For the differences between [1] and our paper, there are 3 aspects: 1) **Confidence Quantifying**. [1] relies on supervisedly trained modules to give confidence, e.g., Longformer-large reading comprehension model pretrained on SQUAD. However, for LLMs, self-calibration is needed. 2) **Decomposition Accurary**. [1] uses large-scale of (question, sub-question) parallel data to supervisedly train a question decomposer with high accuracy. However, for LLMs, question decomposition is conducted with few-shot prompting, which is not that accurate. How to recover from the local wrong decomposition is a unique challenge. 3) **Question Type Coverage**. [1] uses 6 predefined symbolic functions to solve specific types of questions, such as [Verify] and [Count]. The coverage is limited, and some questions cannot be represented by the predefined functions ("In between Atsushi Ogata and Ralpha Smart who graduated from Harvard College?" in HotpotQA). For LLMs, the language understanding ability of LLMs.
>
> ### Decomposing Method
> **Q**: Can some simple semantic parsing method be used to decompose the question? Why is the maximum number of nodes in each layer set to 3? Can different numbers be used for decomposition?
>
> **A**: Semantic parsing based decomposition is not suitable in our framework. Semantic parsing translates questions into logical forms such as SPARQL and lambda-calculus, which is mainly used in KBQA. In our paper, we need natural language to serve as queries for retrieving information from document corpora, and serve as prompt to query LLMs. Therefore, semantic parsing which outputs logical forms is not suitable.
>
> Note that we set the maximum number of child nodes for each node to be 3 (instead of each layer). This is because complex questions can be decomposed into 2 or 3 simper sub-questions, and questions with more than 3 hops can be recursively decomposed (see Figure 2 in our paper for example).
>
> ### Uncertainty Estimation
> **Q1**: Can uncertainty estimation of the intermediate answers be used with other reasoning frameworks, such as sequential decomposition?
>
> **A2**: Yes, Section 5.2.1 and the "SD" in Table 2 in our paper is just the case of sequential decomposition with our uncertainty estimation method.
> The following table contains the performance of sequential decomposition (SD) from our submitted paper:
>
> |          | HotpotQA | MuSiQue | 2WikiMQA |
> |----------|:--------:|:-------:|:--------:|
> | OneR     |   53.2   |  25.7   |   48.1   |
> | IRCoT    |   60.2   |  34.2   |   63.8   |
> | Self-Ask |   49.4   |  33.4   |   66.6   |
> | SD       |   51.7   |  39.0   |   69.2   |
> | ProbTree (ours) |   64.1   |  41.5   |   71.8   |
>
> We can see that SD outperforms Self-Ask on HotpotQA and outperforms all the baselines except ProbTree on MuSiQue and 2WikiMQA, showing that our uncertainty estimation is also useful for other reasoning frameworks.
>
> **Q2**: Does the uncertainty truly reflect the quality of the decomposition and the accuracy of the answers? Case studies or statistics of the consistency between incorrect answers and uncertainty scores are desired.
>
> **A2**: We manually analyze 150 errors (50 per dataset) output by ProbTree and find only a small (~5-15%) number attributed to the uncertainty estimation not selecting the correct answer. The ablation study (lines 476-490) also proves our uncertainty estimation method can better truly reflect the quality of LLM outputs compared with self-consistency.
>
> Actually, before initiating the reasoning framework, we first conducted a pilot study to investigate the uncertainty estimation. We wanted to investigate the incorrect answers and their uncertainty scores. We randomly sampled 500 questions from HotpotQA, and found that the average confidence score for all the 500 examples is ***-0.142***, and the average confidence score for examples with answer F1 ***1.0*** is ***-0.109***, and that of examples with answer F1 ***0.0*** is ***-0.176***. The low confidence of incorrect answers and high score of correct ones indicates the effect of our method.
>
> ### Child-aggregation Reasoning
> **Q**: Why child-aggregation reasoning can generate accurate answers with incorrect sub-question context shown in Table 3?
>
> **A**：Evidence has shown that LLMs possess the ability to recognize what is true or useful in the given context to some extent [2][3].  For example, [2] shows LLMs are calibrated on multiple choice questions and True/False tasks. [3] shows that LLMs have the ability to select useful information from multiple reasoning chains (some chains may be wrong), aggregate them and output the final right answer.
>
> In our case, LLMs can also select useful and relevant information from (sub-question, answer) pairs and based on that further reason. For example, the correct decomposition of ***q0*** should be <***q1***,  ***q2***>. However, the LLM incorrectly decomposes it as <***q1***,  ***q2'***>. For Child-aggregating QA, since we give LLMs the original question ***q0*** as context, it knows that it should find which Canadian rock band had a drummer that was inducted into the Modern Drummer Hall of Fame. ***q2'*** indicates that Rush is the eligible band. Combined with the answer of ***q1***, LLM can give the accurate answer "Rush". Therefore, Child-aggregating QA meta-reasons with information from child nodes, and enables our ProbTree to recover from the wrong decomposition.
>
> |          |Detailed question | Answer |
> |----------|:--------:|:--------:|
> | q0    |   Which Canadian rock band released a song called "Counterparts" and had a drummer who was inducted into the Modern Drummer Hall of Fame?   | Rush |
> | q1    |   Which Canadian rock band released a song called "Counterparts"?   | Rush |
> | q2    |   Which Canadian rock band had a drummer who was inducted into the Modern Drummer Hall of Fame?   | Rush |
> | q2'   |   Which drummer of Rush was inducted into the Modern Drummer Hall of Frame?  | Neil Peart |
>
>
> ### Answer Extraction
> **Q** The process of extracting the intermediate answers for the next step of reasoning is not discussed.
>
> **A**: The output of LLMs is in the format "{explanation} So the answer is: {answer}", so the answer can be extracted programmatically after "So the answer is: ". We will add the details in our revision.
>
>
>
>
> [1] Zhang, Jiajie, et al. "Reasoning over Hierarchical Question Decomposition Tree for Explainable Question Answering." ACL 2023.
>
> [2] Kadavath, Saurav, et al. "Language models (mostly) know what they know." arXiv preprint arXiv:2207.05221, 2022.
>
> [3] Ori, Yoran, et al. "Answering Questions by Meta-Reasoning over Multiple Chains of Thought". arXiv preprint arXiv:2304.13007, 2023.

---

### Official Review · Reviewer_3643 · 2023-08-05

**Soundness:** 4

**Excitement:**

3: Ambivalent: It has merits (e.g., it reports state-of-the-art results, the idea is nice), but there are key weaknesses (e.g., it describes incremental work), and it can significantly benefit from another round of revision. However, I won't object to accepting it if my co-reviewers champion it.

**Paper Topic And Main Contributions:**

The paper utilizes various prompts to answer multi-hop questions (HotpotQA, MusiQue, 2WikiMultihopQA). It first generates subquestions which might form a tree (where answers of some questions feed into other questions), and then answers each subquestion by traversing the tree.

The main innovation over similar previous work is that each subquestion is answered using 3 different strategies, and each strategy is scored and the best one is used. The three strategies are: 1) Child-aggregate: Show child questions and answers in the tree as context, 2) Open-book: Context is retrieve snippets for the subquestions from a corpus, 3) Closed-book: No context (rely on models own knowledge).

Each answer is preceded by an explanation. The main way of scoring each answer is the average log-probability of each token in the explanation. The child-aggregate strategy also includes average token scores for the decomposition and the child answers.

Answer F1 scores are compared with various baselines and existing models, showing general gains, some of them quite substantial. There is also some analysis breaking down performance per reasoning type, ablation of many of the components (leave out one of the strategies, pick random answers rather than higher scoring one). Also, there's a useful error analysis breaking down the types of errors being made, including a small (~5-15%) number attributed to the confidence scoring not selecting the correct answer.

**Questions For The Authors:**

A: It seems strange that the answer tokens themselves are not part of the scoring, is there a reason for this?
B: Could there be a way for the system to recover from decomposition mistakes (e.g., subquestions that cannot be answered)

**Reasons To Accept:**

Shows a potentially useful way of incorporating multiple strategies into a reasoning system. Gets good scores versus comparable systems.

Useful analyses and ablations, giving insights into the performance

**Reasons To Reject:**

Possibly not enough novelty to warrant acceptance.

The scoring based on probabilities per token seems quite fragile, even if it's seemingly effective here. Could have some more analysis of this, and perhaps compared to an approach where the model itself estimates its uncertainty in language.

**Reproducibility:**

4: Could mostly reproduce the results, but there may be some variation because of sample variance or minor variations in their interpretation of the protocol or method.

**Reviewer Confidence:**

4: Quite sure. I tried to check the important points carefully. It's unlikely, though conceivable, that I missed something that should affect my ratings.

---

> ### Author Rebuttal · Authors · 2023-08-28
>
> We are glad to hear that the reviewer finds our reasoning system useful and analyses insightful.
>
> ### Confidence Scoring
> **Q:**
> The scoring based on probabilities per token seems quite fragile, even if it's seemingly effective here. Could have some more analysis of this, and perhaps compared to an approach where the model itself estimates its uncertainty in language.
>
> It seems strange that the answer tokens themselves are not part of the scoring, is there a reason for this?
>
> **A:**
> Thank you for raising this question. Actually, estimating the
> confidence levels associated with the responses of LLMs is an important research topic. In the existing literature, methods can be grouped into three categories:  1) **logits**-based methods that focus on the probabilities derived from model logits; 2) **verbalized** confidence that directly asks a
> model to output its confidence; 3) **consistency**-based methods that yield multiple responses and use consistency as a surrogate for confidence.
>
> Before initiating the reasoning framework, we first conducted a pilot study to investigate whether our explanation logits-based confidence is effective.
> As shown in the following table, we compare different confidence calibration methods. For each method, given a question, we query LLMs multiple times (decode one response with greedy decoding, and sample another four with
> temperature t = 0.7), calculate the confidence score for each response, and take the most confident one as the final answer. Better answer F1 indicates better confidence calibration.
> We randomly sampled 500 questions from HotpotQA, and 500 from MuSiQue to conduct the pilot study. Answer F1 results show that our explanation logits-based confidence outperforms others, including the model itself estimating its uncertainty in language, and taking answer tokens as part of scoring.
>
>
>
> |          | logits (explanation) | logits (explanation+ans) | logits (ans) | verberlize | consistency | vanilla
> |----------|:--------:|:-------:|:--------:| :--------:| :--------:| :--------:|
> | HotpotQA |   47.8  | 47.5   |   43.7 | 44.5 | 47.6 | 47.1
> | MuSiQue  |   26.0  |  25.4  |   24.1 | 22.0 | 25.7 | 25.2
>
>
> ### Recovering from Decomposition Mistakes
>
>
>
> **Q:** Could there be a way for the system to recover from decomposition mistakes (e.g., subquestions that cannot be answered)
>
>
> **A:** Yes. The system can recover from decomposition mistakes. Because for each question, the most confident answer is selected from Closed-book QA (CB), Open-book QA (OB) and Child-aggregating QA (CA). For each QA strategy, LLMs have the chance to output a correct answer even if the decomposition is wrong. The correct answer can be selected as the final answer because its confidence score is often higher compared with the wrong ones. Specifically, CB does not rely on decomposition. OB reasons with the retrieved paragraphs for itself and its descendants. Wrong descendant due to decomposition mistake will cause irrelevant retrieved paragraphs, and OB can recover from it. CA can also
> recover from wrong decompositions because LLMs can extract useful information from child information and output a correct answer (a concrete example is shown in Table 3 in the paper).

---

### Meta-Review · Area_Chair_NWHx · 2023-09-16

**Recommendation:** 4

**Metareview:**

This paper proposes prompting strategies for multi-hop question answering on HotpotQA, MusiQue, and 2WikiMultihopQA. It generates a set of subquestions that forms a tree (i.e., answers to some questions feed into other questions) and then answers each subquestion by traversing the tree. The reviewers found the results convincing and the paper well-written. On the other hand, two reviewers noted similarities between the proposed approach and other approaches inspired by chain-of-thought prompting. Thus, they had lower excitement scores than soundness scores.

---

### Decision · Program_Chairs · 2023-10-07

**Decision:**

Accept-Findings

**Comment:**

This paper proposes prompting strategies for multi-hop question answering on HotpotQA, MusiQue, and 2WikiMultihopQA. It generates a set of subquestions that forms a tree (i.e., answers to some questions feed into other questions) and then answers each subquestion by traversing the tree. The reviewers found the results convincing and the paper well-written. On the other hand, two reviewers noted similarities between the proposed approach and other approaches inspired by chain-of-thought prompting. Thus, they had lower excitement scores than soundness scores.